# Extension of a Monolayer Energy-Budget Degree-Day Model to a Multilayer One

**Julien Augas** [1,*], **Etienne Foulon** [1] , **Alain N. Rousseau** [1] **and Michel Baraër** [2]

1 Institut National de la Recherche Scientifique, Centre Eau Terre Environnement, 490, rue de la Couronne, Québec, QC G1K 9A9, Canada; etienne.foulon@inrs.ca (E.F.); alain.rousseau@inrs.ca (A.N.R.)
2 Département de Génie de la Construction, École de Technologie Supérieure, Montréal, QC H3C 1K3, Canada
* Correspondence: julien.augas@inrs.ca

**Abstract:** This paper presents the extension of the monolayer snow model of a semi-distributed hydrological model (HYDROTEL) to a multilayer model that considers snow to be a combination of ice and air, while accounting for freezing rain. For two stations in Yukon and one station in northern Quebec, Canada, the multilayer model achieves high performances during calibration periods yet similar to the those of the monolayer model, with KGEs of up to 0.9. However, it increases the KGE values by up to 0.2 during the validation periods. The multilayer model provides more accurate estimations of maximum *SWE* and total spring snowmelt dates. This is due to its increased sensitivity to thermal atmospheric conditions. Although the multilayer model improves the estimation of snow heights overall, it exhibits excessive snow densities during spring snowmelt. Future research should aim to refine the representation of snow densities to enhance the accuracy of the multilayer model. Nevertheless, this model has the potential to improve the simulation of spring snowmelt, addressing a common limitation of the monolayer model.

**Keywords:** multilayer structure; snow water equivalent; ice/air mixture; snow modeling; snowmelt; sensitivity analysis; snow height; winter snow peak





## 1. Introduction

Understanding the hydrological cycle is a paramount challenge for humanity, as it is essential for protecting against floods, mitigating droughts, meeting water needs for industrial and domestic purposes, and informing weather and climate predictions. Within this cycle, one crucial component is snowfall. Although in the Northern Hemisphere, snow typically constitutes around 6–10% of total precipitation, it can exceed 50% in specific regions [1]. Accumulating as a heat-deficient solid water reservoir, snowpacks experience rapid spring melting, leading to distinctive seasonal flooding patterns. Notably, snowmelt has been found to contribute substantially to annual streamflow in various geographic contexts. For example, in Indian glacier-fed basins, snowmelt accounts for 27–44% [2], and in Czech Republic watersheds 17–42% [3]. Meanwhile, snowmelt can play a pivotal role in groundwater recharge. For example, in the Nelson River Basin, Canada, Jasechko et al. [4] determined that the fraction of precipitation recharging aquifers is 1.3 to 5 times higher during cold months, with negative mean monthly temperatures, than during warmer months. Snow can pose a challenge in mountainous areas like the Andes [5] or Iran [6], and snowmelt remains a concern. Consequently, accurate modeling of snow cover becomes crucial for streamflow modeling.

Snow–water equivalent (or *SWE*) represents one of the key physical characteristics and is defined as the depth of water on the ground if the snow were in a liquid state. For hydrological models simulating water transfers within the hydrological cycle, *SWE* represents an essential variable and is equivalent to the product of snowpack height and snow density (mass of snow per unit volume of snowpack). Another key characteristic, albedo, represents

the proportion of solar radiation reflected by the snowpack surface and thus directly affects the amount of absorbed solar energy. In the case of fresh snow, assuming reflectivity is isotropic, its specular component—which entails unidirectional reflection—strengthens as the snowpack ages and undergoes repeated melting and recrystallization events [7], affecting snow metamorphosis and sublimation [8]. Finally, the temperature or calorific deficit of the snowpack assists in determining the snow maturation process and proximity to melting.

Given our physical understanding of snow, multiple snow models have been developed to tackle specific issues related to water resource management, including integrated management, avalanche prediction, climate studies, infrastructure planning, environmental impact, or even scientific research in fields such as ecology or glaciology. As is typical with modeling, the complexity of those models is tailored to the objective they seek to address. Table A1 in Appendix A presents a selection of snow models that differ in their design approach, consideration of the simulated phenomena given the available data, and thus representation of the snowpack structure.

Without delving into the details of each snow model, which would go beyond the scope of this paper, Table A1 highlights a major difference in complexity between monolayer models, which are all daily models and consider only a limited number of phenomena, and multilayer ones that can provide snow cover modeling at 10 min intervals and consider a wider range of phenomena. The snow model of HYDROTEL stands out among monolayer models as the most advanced in terms of physical representation. It encompasses phenomena found in simple models like CEMANEIGE and HBV (i.e., snow accumulation and melting), as well as numerous phenomena typically associated with multilayer modeling approaches (e.g., convective heat, precipitation heat, soil heat, compaction, mixing, radiation heat/melting degree-day, water retention). HYDROTEL [9,10] is the semi-distributed hydrological model at the core of operational hydrologic forecasting systems in the Quebec [11,12], Yukon [13–15], and Southern Québec Hydroclimatic Atlas [16–18], as well as in several other studies, such as on the effect of global warming on environmental flows [19–21] or the role of wetlands on mitigating floods and droughts [22–24]. These applications are made in a Canadian context, where most watersheds are subject to significant snowfall. In these regions, spring freshets often result in annual streamflow peaks, sometimes accompanied by rain-on-snow events [25], which can augment lateral outflows and impede soil infiltration [26]. As a result, accurate simulation of snowmelt becomes critical for effectively predicting streamflow, accounting for both surface runoff and groundwater recharge [27]. One notable feature of HYDROTEL's snow model is its consideration of snow as a monolayer structure [28]. However, the literature suggests that adopting a multilayer representation of the snowpack can significantly improve *SWE* dynamics. For instance, Saha et al. [29] demonstrated substantial enhancements in snowpack height and *SWE* estimations with the use of the six-layer Noah model compared to its conventional monolayer version. In addition, Domine et al. [30] highlighted the significance of accurately modeling the thermal properties of snow for estimating soil water mass balance, suggesting that a multilayer structure can effectively capture density profiles and improve the representation of thermal characteristics. In addition to incorporating a multilayer structure, some models treat snow as a heterogeneous material, accounting for the proportions of air, ice, and water in the snow cover. For example, the SNOWPACK model [31] considers these factors, while the GEOTOP [32] and SeNORGE [33] models represent snow as a mixture of solid and liquid water. Furthermore, the integration of freezing rain enables the direct formation of an ice layer over an existing snow cover, as observed in studies by Henson et al. [34] and Quéno et al. [35].

These different considerations offer potential directions of improvement for snow modeling in HYDROTEL. Given the advancements in modeling sophistication and computational capabilities, this paper focuses on developing a multilayer version of the hybrid energy balance/degree-day snow model of HYDROTEL, assuming snowpack is predominantly composed of ice with interspersed air. As ice exhibits distinct thermal properties

compared to those of air, this development impacts heat transfer between layers while creating discontinuities in the physical properties and ensuing temperature and density profiles. This development aligns with the parsimonious structure of HYDROTEL, which has positioned the model as a robust model in Canada. The end goal is not to transform HYDROTEL into a complex and computationally intensive model but rather to assess the potential improvements associated with using a multilayer structure within a relatively simple, physics-based, semi-distributed model.

This paper is organized as follows. First, we describe the original snow model design of HYDROTEL, detailing the modifications from a monolayer model to a multilayer one, and present a sensitivity analysis of the additional parameters. The mono- and multilayer models are then calibrated based on *SWE*, and the resulting differences are highlighted. The modeling is validated using Gamma MONitor (GMON) stations in the Necopastic watershed (Quebec), Lower Fantail, and Wheaton (Yukon) River basins. The effects of the model design on energy balance dynamics, state variables, and characteristic dates of the snowpack are analyzed, followed by a discussion and conclusion.

## 2. Materials and Methods

### 2.1. Core Equations of the Monolayer Snow Model

This section presents the governing equations of the monolayer snow model of HYDROTEL [28] focusing on modeling the phenomena introduced in Table A1, namely, snow accumulation, advected heat transfer from precipitation, soil heat transfer, snow compaction, snow water content, and construction of the thermal energy budget (through blending net short-wave radiation and degree-day concepts).

The operation of the model is parsimonious and only requires three input variables, that is, daily total precipitation, and minimum and maximum air temperatures. The model is physics-based, using degree-day equations while building a thermal energy budget based on the heat deficit of a monolayer snowpack. This budget is as follows (see Appendix B for a detailed mathematical description of each term):

$$\frac{\Delta U}{\Delta t} = u_r + u_c + u_{s-s} + u_{a-s} + u_{ac} - u_s \tag{1}$$

where $\frac{\Delta U}{\Delta t}$ is the daily rate of change in the snowpack heat deficit (J.m$^{-2}$.s$^{-1}$); $u_r$, $u_c$, $u_{s-s}$, $u_{a-s}$, and $u_{ac}$ are decreases in heat deficits due to rainfall, conduction, transfer from the soil (at the snow–soil interface), net radiation (at the air–snow interface), and from the water retained on the previous day, respectively; and $u_s$ is the increase in heat deficit due to solid precipitation.

The energy assessment is applied to a snow layer. Liquid and solid precipitations are derived from total precipitation, daily minimum and maximum air temperatures, and a temperature threshold. When the air temperature is sufficiently cold (below the threshold), all precipitation falls as snow (Equation (2a)), whereas when the temperature is warm enough (above the threshold), it falls as rain (Equation (2b)). In between, total precipitation results in a mix of snow and rain (Equation (2c)).

$$R = 0; S = P_t \; if \; T_{max} \leq T_s \tag{2a}$$

$$R = P_t; S = 0 \; if \; T_{min} > T_s, \tag{2b}$$

$$R = P_t\left(\frac{T_{max} - T_s}{T_{max} - T_{min}}\right); S = P_t\left(\frac{T_s - T_{min}}{T_{max} - T_{min}}\right) \; otherwise \tag{2c}$$

where $R$, $S$, and $P_t$ are liquid, solid, and total daily precipitation rates (m.s$^{-1}$), respectively; $T_{max}$ and $T_{min}$ are the maximum and minimum daily air temperatures, respectively; and $T_s$ is the temperature threshold.

The density of falling snow is computed as follows:

$$\rho_s = 151 + 10.63 \left( \frac{T_{max} + T_{min}}{2} \right) + 0.2767 \left( \frac{T_{max} + T_{min}}{2} \right)^2 \text{ if } \frac{T_{max} + T_{min}}{2} \geq -17 \quad (3a)$$

$$\rho_s = 50 \text{ } if \text{ } \frac{T_{max} + T_{min}}{2} < -17 \quad (3b)$$

where $\rho_s$ is the density of fresh snowfall (kg.m$^{-3}$), and $T_{max}$ and $T_{min}$ are the maximum and minimum daily air temperatures, respectively.

The snowpack is subject to compression, and a reduction in height ($Sett$) is estimated using Equation (4). Thus, $Sett$ is subtracted from the current height of the snow layer. When negative, $Sett$ is set to 0.

$$Sett = H \, Set_{Coef} \left( 1 - \frac{\rho_{snow}}{\rho_{max}} \right) \quad (4)$$

where $Sett$ is snowpack height lost to compaction (m), $H$ is the snow height (m), $Set_{Coef}$ is the compaction coefficient ($-$), and $\rho_{max}$ is the maximum achievable density (kg.m$^{-3}$).

When the total snowpack heat deficit is replenished, a potential snow melt is computed from the excess heat, triggering a phase change as per Equation (5).

$$PM = \frac{\Delta U_{tot}}{C_f \, \rho_w} \quad (5)$$

where $PM$ is the resulting amount of $SWE$ undergoing a phase change (m), $\Delta U_{tot}$ is the total heat deficit (J.m$^{-2}$), $\rho_w$ is the liquid water density (1000 kg.m$^{-3}$), and $C_f$ is the latent heat of the fusion of water (335,000 J.kg$^{-1}$).

The maximum water retention capacity ($RC_{max}$) is computed as follows:

$$RC_{max} = 0.1 \frac{\rho_{snow}}{\rho_w} SWE \quad (6)$$

where $RC_{max}$ is the maximum snow cover capacity of water retention (m), and $SWE$ is the snow water equivalent following the removal of $PM$ (m).

The actual snowmelt ($AR$) is computed as the difference between potential melt and $RC_{max}$ (Equations (7a) and (7b)).

$$AR = \frac{PM}{\Delta t} \text{ if } PM \leq RC_{max} \text{ then } AM = 0 \quad (7a)$$

$$AR = \frac{RC_{max}}{\Delta t} \text{ and } AM = \frac{PM - RC_{max}}{\Delta t} \text{ otherwise} \quad (7b)$$

where $AM$ is the actual snowmelt (m.s$^{-1}$), $AR$ is the actual retention, and $\Delta t$ is the computational time step.

Finally, the snowpack mass balance is the sum of the snowfall and rainfall when there is snow on the ground; otherwise, rainfall either percolates or runs off.

$$\frac{\Delta SWE}{\Delta t} = R + S - AM \quad (8)$$

### 2.2. Extension of the Monolayer Snow Model

Several modifications to the model are considered, including a change from a monolayer to a multilayer structure. Additionally, some variables are estimated by considering snow as a material composed of both ice and air. Furthermore, freezing rain is made possible given its potential to alter the heat transfer inertia between each layer. Finally, some modifications are introduced to the equations describing snow compression and maximum water retention capacity to account for the changes. These modifications are described in the next subsections.

### 2.2.1. A Multilayered Structure

Any snowfall in the absence of snow on the ground or above a pure ice layer (layer with a density of 917 kg.m$^{-3}$) leads to the creation of a new layer with a specific mass and heat deficit. If these criteria are not met, and if the snowfall water equivalent is less than a threshold value *St*, then the incoming mass and heat deficit are incorporated into the current layer at the air–snow interface. Otherwise, a new layer is established, as illustrated in Figure 1. *St* serves as a calibration parameter, allowing for concurrent optimization of energy transfers and restricting the number of layers. Some energy transfer processes solely affect specific layers. For instance, heat input from the ground solely influences the layer at the ground–snow interface, while radiation exclusively warms up the layer at the air–snow interface. For the latter layer of the monolayer model, heat loss through conduction and heat gain via radiation are enabled when the air temperature is below or above the melting threshold temperature $T_0$, respectively. Furthermore, in instances where melting exceeds the water retention capacity, excess water seeps into the underlying layer at a temperature of 0 °C. Excess heat is used for phase change; if the uppermost snow layer has undergone a phase change, any residual heat is then transferred downwards. Consequently, the energy balance can be expressed using Equations (9a)–(9c) for the top layer, any intermediate layers, and the bottom layer, respectively.

$$\frac{\Delta U_k}{\Delta t} = u_r + u_c + u_{a-s} + u_{ac} - u_s \tag{9a}$$

$$\frac{\Delta U_k}{\Delta t} = u_c + u_{ac} + u_{ex,k+1} + u_{perc,k+1} - u_{ex,k} - u_{perc,k} \tag{9b}$$

$$\frac{\Delta U_1}{\Delta t} = u_c + u_{s-s} + u_{ac} + u_{ex,2} + u_{perc,2} \tag{9c}$$

where $u_{ex}$ is the excess heat from melting in the upper layer or the heat transfer due to phase change of freezing rain from the upper layer (more detail below in the article) (J.m$^{-2}$.s$^{-1}$), $u_{perc}$ is the heat variation due to infiltration from the upper layer (J.m$^{-2}$.s$^{-1}$), and $k$ stands for the $k^{th}$ snow layer from the ground surface.

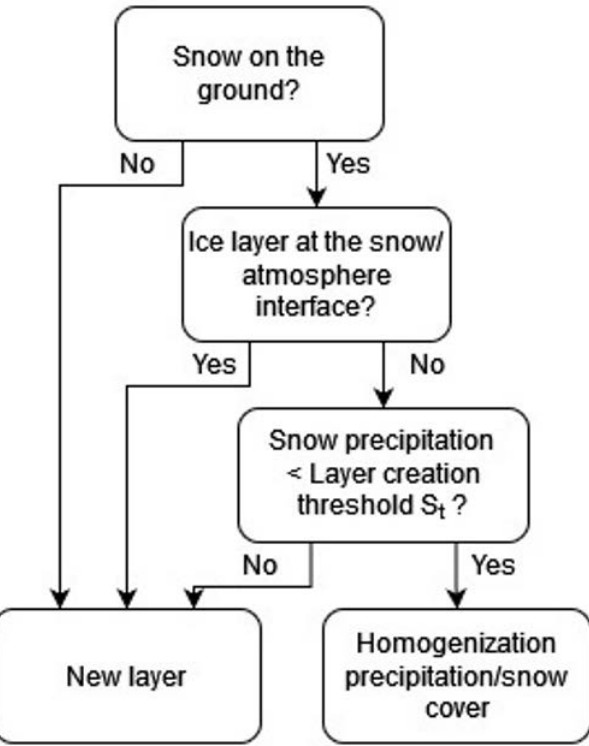

**Figure 1.** Snow layer creation scheme for the proposed multilayer model.

The heat input from percolation, $u_{perc}$, is evaluated in a similar manner to $u_r$. In both cases, the heat input to the snowpack comprises the cumulative sensible heat loss of the liquid water lowered to $0\,°C$, the ensuing latent heat of fusion (phase change), and the heat released to adjust the new ice crystals to the snowpack temperature. They are described by Equations (10a) and (10b) for the modification of thermal energy from rainfall and percolation, respectively. The rainfall occurred on the top layer, which is noted as $k'$ below.

$$\begin{cases} u_r = \rho_w R \left( C_w T_m + C_f \right) \left( 1 - \frac{R}{SWE_{k'} + R} \right) + \frac{R U_{k'}}{SWE_{k'} + R} \text{ if } T_m > 0 \\ u_r = \rho_w R \left( C_s T_m + C_f \left( 1 - \frac{R}{SWE_{k'} + R} \right) \right) + \frac{R U_{k'}}{SWE_{k'} + R} \text{ otherwise} \end{cases} \tag{10a}$$

$$u_{perc,k} = \rho_w Ru_{k+1} C_f \left( 1 - \frac{Ru_{k+1}}{SWE_k + Ru_{k+1}} \right) + \frac{Ru_{k+1} U_k}{SWE_k + Ru_{k+1}} \tag{10b}$$

where $C_w$ and $C_s$ are specific heat capacities of water and snow (4184 $J.kg^{-1}.°C^{-1}$ and 2093.4 $J.kg^{-1}.°C^{-1}$), respectively; $C_f$ is the heat of fusion of water (335,000 $J.kg^{-1}$); $R$ is the rainfall rate ($m.s^{-1}$); $Ru_{k+1}$ is the percolation rate of the $k+1^{th}$ layer ($m.s^{-1}$); $T_m$ is the mean air temperature ($°C$); $SWE_k$ is the snow water equivalent (m); and $U_k$ is the heat deficit of the $k$th layer.

### 2.2.2. Snow as a Medium of Ice and Air

The snowpack is regarded as a medium comprising different constituents whereby the properties and proportions of each component contributes to the estimation of various snow characteristics. Appendix E describes how the volumetric proportions of air and ice are estimated, assuming liquid water constitutes a non-significant portion of the snowpack during winter. This assumption is based on observations made by Koch et al. [36], where the volumetric liquid water content peaked at a maximum of 8% at the end of the melting phase or during instances of liquid precipitation. This is consistent with the assumption that liquid water in the original snow model is entirely frozen at the daily time step. Leveraging the relationship derived for snow density (Appendix E) and the linear correlation proposed by Evans [37] to gauge the relative dielectric permittivity of snow from those of ice and air, all snow layer characteristics are determined based on the proportions of ice and air. For heat loss by conduction, the thermal diffusivity of snow is computed for each layer using Equation (11).

$$D_{s,k} = \left( \frac{\rho_{s,k} - \rho_{a,k}}{\rho_i - \rho_{a,k}} \right) D_{i,k} + \left( \frac{\rho_i - \rho_{s,k}}{\rho_i - \rho_{a,k}} \right) D_{a,k} \tag{11}$$

where $D_{s,k}$ is the snow diffusivity ($m^2.s^{-1}$); $\rho_{s,k}$, $\rho_{a,k}$, and $\rho_i$ are the snow, air, and ice densities ($kg.m^{-3}$), respectively; $D_{i,k}$ and $D_{a,k}$ are the ice and air thermal diffusivities ($m^2.s^{-1}$), respectively; and $k$ stands for the $k$th snow layer.

The thermal diffusivities of ice and air are computed using Equation (12):

$$D_{m,k} = \frac{K_{m,k}}{\rho_{m,k} C_{s,m,k}} \tag{12}$$

where $D_{m,k}$ is the thermal diffusivity of the $k$th snow layer made of a material $m$ ($m^2.s^{-1}$), $K_{m,k}$ is the thermal conductivity ($W.m^{-1}.°C^{-1}$), $\rho_{m,k}$ is the density ($kg.m^{-3}$), and $C_{s,m,k}$ is the specific heat ($J.kg^{-1}.°C^{-1}$).

Estimates of the thermal conductivities of ice [38] and air [39] are derived from Equations (13) and (14), respectively.

$$K_{i,k} = 1.16\left(1.91 - 8.66.\,10^{-3}\,T_k + 2.97.10^{-5}\,T_k^2\right) \tag{13}$$

where $T_k$ is the temperature of the $k$th layer ($°C$).

$$K_{a,k} = 1.5207.10^{-11}(273.15 + T_k)^3 - 4.857.10^{-8}(273.15 + T_k)^2$$
$$+1.0184.10^{-4}(273.15 + T_k) - 3.9333.10^{-4} \tag{14}$$

$T_k$ is a function of the total heat deficit $\Delta U_{tot,k}$ computed for the $k^{th}$ layer using Equation (15):

$$T_k = \frac{\Delta U_{tot,k}}{SWE_k \, C_s \, \rho_w} \tag{15}$$

For ice, the density and specific heat are deemed constant for any temperature and are set at 917 kg.m$^{-3}$ and 2093.4 J.kg$^{-1}$.°C$^{-1}$, respectively. For air, the density (ideal gas law under normal pressure conditions) and specific heat [39] are computed using Equations (16) and (17), respectively:

$$\rho_{a,k} = 1.292 \, \frac{273.15}{273.15 + T_k} \tag{16}$$

$$C_{s,a,k} = 1.9327.10^{-10}(273.15 + T_k)^4 - 7.999.10^{-7}(273.15 + T_k)^3$$
$$+1.1407.10^{-3}(273.15 + T_k)^2 - 4.489.10^{-1}(273.15 + T_k) \tag{17}$$
$$+1.0575.10^3$$

where $T_k$ stands for the temperature of the $k^{th}$ layer (°C).

Snow albedo is determined by snow grain metamorphism, which also causes the snowpack to become denser. However, our snow model assesses albedo based on snow density since snow grain size and shape are not evaluated. Here, snow albedo is estimated based on the proportion of ice and air in the surface layer. This approach is reminiscent of the optical paths of radiation that are absorbed by ice crystals instead of being reflected or transmitted through them. Nevertheless, since the albedo of air cannot be defined, fresh snow was employed as a surrogate material. Indeed, fresh snow constitutes a blend of ice and air with a very high porosity.

Perovich et al. [40] measured an ice albedo of 0.5 in the Arctic for snow on a frozen pothole. The albedo of fresh snow is 0.9 [41] for a 50 kg.m$^{-3}$ density, which is consistent with that of snowfall computed in the monolayer mode. The albedo of snow as a composite material is thus computed using Equation (18):

$$\alpha_s = \left(\frac{\rho_s - \rho_{fs}}{\rho_i - \rho_{fs}}\right)\alpha_i + \left(\frac{\rho_i - \rho_s}{\rho_i - \rho_{fs}}\right)\alpha_{fs} \tag{18}$$

where $\alpha_i$ and $\alpha_{fs}$ are albedos of ice (0.5) and fresh snow (0.9), respectively, and $\rho_{fs}$ is the fresh snow density (50 kg.m$^{-3}$).

### 2.2.3. Freezing Rain

Freezing rain occurs upon contact with surfaces when raindrops become supercooled while passing through a freezing layer of air. It is characterized by a heat deficit due to changes in both phase and air temperature. Like how the monolayer model manages precipitation that freezes within the snow cover, the freezing rain heat deficit from the newly created layer is computed using Equation (19):

$$u_s = \rho_w \left(C_f - C_w \frac{T_{max} + T_{min}}{2}\right) R \tag{19}$$

where $\rho_w$ is the liquid water density (1000 kg.m$^{-3}$); $C_f$ is the heat of fusion of water (335,000 J.kg$^{-1}$); $C_w$ is the specific heat capacity of water (4184 J.kg$^{-1}$.°C$^{-1}$); $T_{max}$ and $T_{min}$ are the maximum and minimum daily air temperatures, respectively (°C); and $R$ is the daily liquid precipitation rate (m.s$^{-1}$).

Ice is a better heat conductor than air—about 100 times more, according to Equations (13) and (14). That is why upon freezing, the excess heat from the phase change is transferred to the snowpack (see Equation (20)). The ice-layer temperature subsequently impacts the conduction heat loss of the lower layer.

$$u_{ex} = \rho_w C_f R \tag{20}$$

where $\rho_w$ is the liquid water density (1000 kg.m$^{-3}$), $C_f$ is the heat of fusion of water (335,000 J.kg$^{-1}$), and $R$ is the daily liquid precipitation rate (m.s$^{-1}$).

It is noteworthy that in the original monolayer model, the cooling of ice from 0 °C down to the snow layer temperature was neglected. This oversight stands corrected in the multilayer model.

### 2.2.4. Compression

Snow is made of ice crystals and can undergo compression due to its own weight. Throughout this process, there is no melting or loss of mass, and the snow is contained within a time-dependent volume, as the bonds between ice crystals strengthen, resulting in a structure that can better withstand gravitational force. For this purpose, compaction is computed using Equation (4), with a distinct maximum density $\rho_{max,l}$.

### 2.2.5. Maximum Water Retention Capacity

Some snow models, such as MASiN [42], estimate the maximum water retention capacity of a layer as a proportion of the volume of air that can retain the melted snow. Since the volume of air is now a variable in the proposed model (see the Section 2.2), this capacity can be computed as follows:

$$RC_{max,k} = \%air \, \frac{\rho_i - \rho_{s,k}}{\rho_i - \rho_{a,k}} \, H_k \tag{21}$$

where $RC_{max,k}$ is the maximum water retention capacity of the $k^{th}$ layer (m), $\%air$ is the ratio of the volume of air that can be filled in by water ($-$), and $H_k$ is the height of the $k^{th}$ layer after melting (m).

Table 1 displays the calibration parameters and their respective physical ranges considered for the two versions of the snow model. They align with typical values employed in HYDROTEL. However, the lower limit of parameter $T_0$ is relatively small, intended for an open vegetation environment. Despite the low probability of reaching this value during the calibration of the hydrological model, it was retained to evaluate the behavior of the snow model should an optimal solution be identified using such a value.

**Table 1.** Snow model calibration parameters.

| Parameter | Model | Meaning | Lower Threshold | Upper Threshold |
|---|---|---|---|---|
| $\rho_{max}$ | Original | Maximum snow density (kg.m$^{-3}$) | 250 | 550 |
| $T_0$ | Original/multilayer | Temperature threshold for net radiation heat gain (°C) | $-8$ | 3 |
| $T_s$ | Original/multilayer | Precipitation separation temperature (°C) | $-1$ | 3 |
| $Set_{Coef}$ | Original/multilayer | Settling coefficient ($-$) | 0.0001 | 0.1 |
| $MR_{a-s}$ | Original/multilayer | Melt rate at air–snow interface (m.day$^{-1}$.°C$^{-1}$) | 0.001 | 0.04 |
| $MR_{s-s}$ | Original/multilayer | Melt rate at snow–ground interface (m.day$^{-1}$) | 0.0001 | 0.002 |
| $S_t$ | Multilayer | New-layer snow precipitation threshold (m.day$^{-1}$) | 0 | 0.06 |
| $\rho_{max,l}$ | Multilayer | Settling maximum snow-layer density (kg.m$^{-3}$) | 350 | 750 |
| $\%air$ | Multilayer | Ratio of the volume of air that can be filled in by water ($-$) | 0.05 | 0.15 |

It is noteworthy that the multilayer snow model introduces three additional calibration parameters while removing one, keeping it relatively parsimonious while allowing for the integration of one new phenomenon: freezing rain.

### 2.3. Framework for Evaluating Different Versions of the Snow Model

The models were calibrated using OSTRICH [43], which provides a choice of different deterministic algorithms, such as steepest descent [44] or multi-start GML with trajectory repulsion [45], as well as stochastic algorithms such as dynamically dimensioned search (DDS) [46] or shuffled complex evolution [47,48]. For this study, we used DDS following the guidelines proposed by Tolson et al. [46]. For the mono- and multilayer versions of the snow model, there are six calibration parameters, requiring at least 18 calibration repetitions (trials) of 100 iterations each.

The Kling–Gupta efficiency (KGE) was used as the objective function [49]:

$$\text{KGE} = 1 - \left[ (1 - \mu_s/\mu_o)^2 + (1 - \sigma_s/\sigma_o)^2 + (1 - r)^2 \right]^{1/2} \tag{22}$$

where $\mu_s$ et $\mu_o$ are the simulated and observed *SWE* averages, respectively; $\sigma_X$ is the standard deviation; and $r$ is the Pearson correlation coefficient.

We conducted a sensitivity analysis using the variogram analysis of response surface (VARS) toolbox from Razavi et al. [50]. Among the various suggested tools, the STAR-VARS method [51], based on a "star-based" sampling strategy, was retained because it is an efficient global sensitivity analysis (GSA) technique for analyzing the variograms of the model. A variogram characterizes the model's spatial covariance structure and takes the following form:

$$\gamma(\vec{h}) = \frac{1}{2|N(\vec{h})|} \sum_{(i,j) \in N(\vec{h})} (y(\vec{x}^A) - y(\vec{x}^B))^2 \tag{23}$$

where $\vec{h}$ is the distance (or direction) between the parameter sets $\vec{x}^A$ and $\vec{x}^B$ in the factor space, $N(\vec{h})$ is the number of pairs of points in the factor space with a distance $\vec{h}$ between them, and $y(\vec{x}^A)$ and $y(\vec{x}^B)$ are the response of the model in the parametric space at locations $\vec{x}^A$ and $\vec{x}^B$, respectively.

Therefore, an increase in the variogram in a direction $\vec{h}$ in the factor space implies a greater variation on $\vec{h}$, indicating a higher sensitivity of the model in this direction.

To combine the various variograms for each parameter, a sensitivity index (IVAR) is generated for each one of them, which integrates the variograms over a scale interval from 0 to $H_i$ for a parameter $i$:

$$IVAR_i(H_i) = \int_0^{H_i} \gamma(h_i) dh_i \tag{24}$$

Based on the recommendation of Razavi and Gupta [52], we calculated the sensitivity index for 50% of the interval ($IVAR_i(0.5)$), corresponding to a scale of $H_i = 0.5$. To facilitate parameter comparison, a relative sensitivity index ($IVAR_{i,50n}$) is estimated for each parameter $i$ as follows:

$$IVAR_{i,50n} = \frac{IVAR_i(0.5)}{\sum_{j=1}^n IVAR_j(0.5)} \tag{25}$$

A temporal sensitivity analysis was performed by estimating the $IVAR_{i,50n}$ for each day using a generalized global sensitivity matrix approach (or GGSM) instead of the previously employed GSA method.

The Latin hypercube sampling method was adopted to generate the parameter sets, using a sampling of parameter sets based on 50 stars with a resolution of 0.1. The time

frame aligns with each period of accessible data, which will be elaborated upon in the case study section.

Two calibration strategies were evaluated to optimize the information obtained from the different datasets of the *SWE* gauge stations presented below. The first strategy was to test the prediction ability of both the monolayer model and the multilayer model. Various calibrations were performed by extracting one year of the datasets for validation, while the remaining years were used for calibration. All possible permutations were evaluated. The second strategy involved using the complete dataset to compare the overall performance of each model. The top ten KGE performances, assessed on *SWE* during the calibration period (or as the optimal compromise between calibration and validation periods for the first strategy), were compared for both models at every *SWE* gauge station. A Wilcoxon rank sum test was performed to compare the median of these performances at each station. A *p*-value of less than 0.05 indicated a significant difference at a 5% type-I error rate. The second strategy consisted of using all available data for calibration with the KGE. In addition to the KGE, the root mean squared error (RMSE) and Nash–Sutcliffe Efficiency (NSE) [53] were computed. For the remainder of this paper, the monolayer model is referred to as "Mo", while the multilayer model is referred to as "Multi".

$$RMSE = \sqrt{\frac{\sum_{i=1}^{n}(SWE_{o,i} - SWE_{s,i})^2}{n}} \tag{26}$$

where $n$ is the number of daily time steps, and $SWE_{o,i}$ and $SWE_{s,i}$ are the observed and simulated *SWE* for day $i$ (m), respectively.

$$NSE = 1 - \frac{\sum_{i=1}^{n}(SWE_{s,i} - SWE_{o,i})^2}{\sum_{i=1}^{n}(SWE_{o,i} - \overline{SWE_o})^2} \tag{27}$$

where $\overline{SWE_o}$ is the mean observed *SWE* over the entire dataset.

Finally, to further substantiate differences between the Mo and Multi models, the snowpack onset and end dates as well as the date of maximum *SWE* and height were compared on an annual basis. The results are presented relative to their absolute seasonal deviations for each set of parameters using Equation (28). The median results are then compared between models at each *SWE* station.

$$A_c = \sqrt{(C_{k,m} - C_{k,o})^2} \ or \ A_c = 100\frac{\sqrt{(C_{k,m} - C_{k,o})^2}}{C_{k,o}} \ (SWE \ max, \ in \ \%) \tag{28}$$

where $A_c$ is the mean value of characteristic $C$, and $m$ stands for the tested model (Mo or Multi) and $o$ the observations for year $k$.

*2.4. Case Study*

Three *SWE* stations were selected for this study based on their differences in altitude and climate. As shown in Figure 2, they are in two distinct regions of Canada. The first *SWE* station (a.k.a. GMON station) and meteorological stations are in the Necopastic River watershed, in a subboreal climate. It is in a 50 m-radius forest clearing, surrounded by a 7 to 8 m-tall spruce trees, with vegetation reaching 3 to 4 m beyond 30 m. The exact altitude of the station is uncertain, but the altitudes of the watershed are between 100 and 180 m. The observed data used in this study were taken from Oreiller et al. [54]. The two other *SWE* stations are in the Upper Yukon River watershed, namely, the Lower Fantail and the Wheaton stations [55]. The Lower Fantail stations are located on an outcrop surrounded by a wetland, at the bottom of a river valley, while the Wheaton stations are located on a ridge crest close to a glacier, surrounded partially by subalpine firs and shrubs. These stations are located in the alpine, subalpine, and boreal eco-climatic regions of the Northern

and Central Cordillera [56]. The exact station altitude is uncertain, but the altitudes of this watershed fall between 640 and 2010 m.

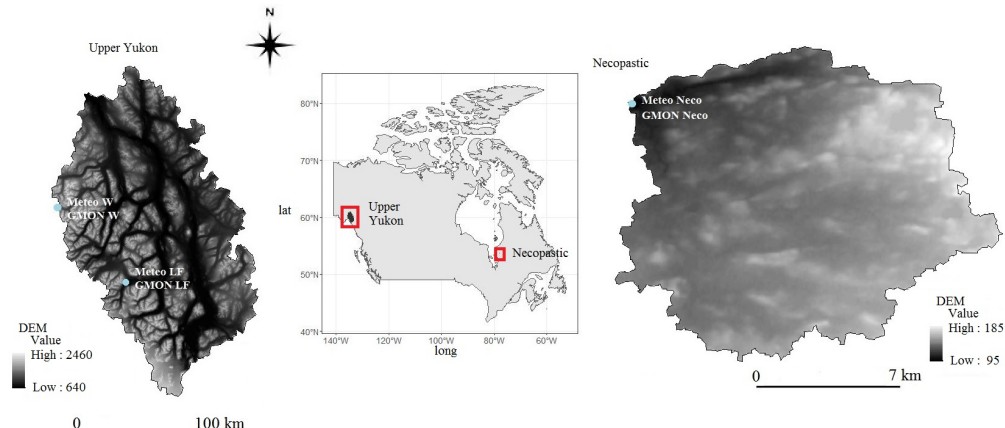

**Figure 2.** Locations of the upper Yukon (**left**) and Necopastic (**right**) watersheds in Canada. Weather and ground snow stations are in blue circles. LF stands for Lower Fantail, W for Wheaton, and Neco for Necopastic.

The weather and *SWE* station metadata are provided in Table 2. During the study, ground-based precipitation measurements were non-continuous in Yukon. Given these conditions, the precipitation times series for modeling was assumed to be the daily increase in observed water equivalents due to the lack of information about wind-related snow transport. The modeled *SWE* was compared to data from the GMON stations, which measure gamma rays naturally emitted by the Earth and attenuated by the snowpack. The measuring principle, developed by Choquette et al. [57], converts gamma radiation measurements into *SWE* (mm). The station sensors at Necopastic and Upper Yukon are GMON3 [58] and CS275s [59], respectively, with measurement uncertainties ranging from $\pm 15$ mm (for *SWE* less than 300 mm) to $\pm 15\%$ otherwise. Figure 3a–c depict precipitation, average air temperature, and *SWE* time series at the three stations. The evaluation of model performance excluded days without *SWE* data.

**Table 2.** Weather and GMON station metadata. Data for the Necopastic watershed are from Oreiller et al. [54]; Upper Yukon data were provided by Yukon Energy.

| Station | Code | Period | Temporal Resolution | Type | Basin |
|---|---|---|---|---|---|
| Necopastic | Meteo_Neco GMON Neco | 2006–2011 | Daily and hourly 6 h | Auto | Necopastic |
| Lower Fantail | Meteo_LF GMON LF | 2014–2017 | Daily and hourly 6 h | Auto | Upper Yukon |
| Wheaton | Meteo_W GMON W | 2014–2017 | Daily and hourly 6 h | Auto | Upper Yukon |

Table 3 shows the average temperature and cumulative precipitation for each hydrological year. The fifth year of the Necopastic station appears to be an aberration. However, the dataset for that year did not account for the summer temperature or precipitation.

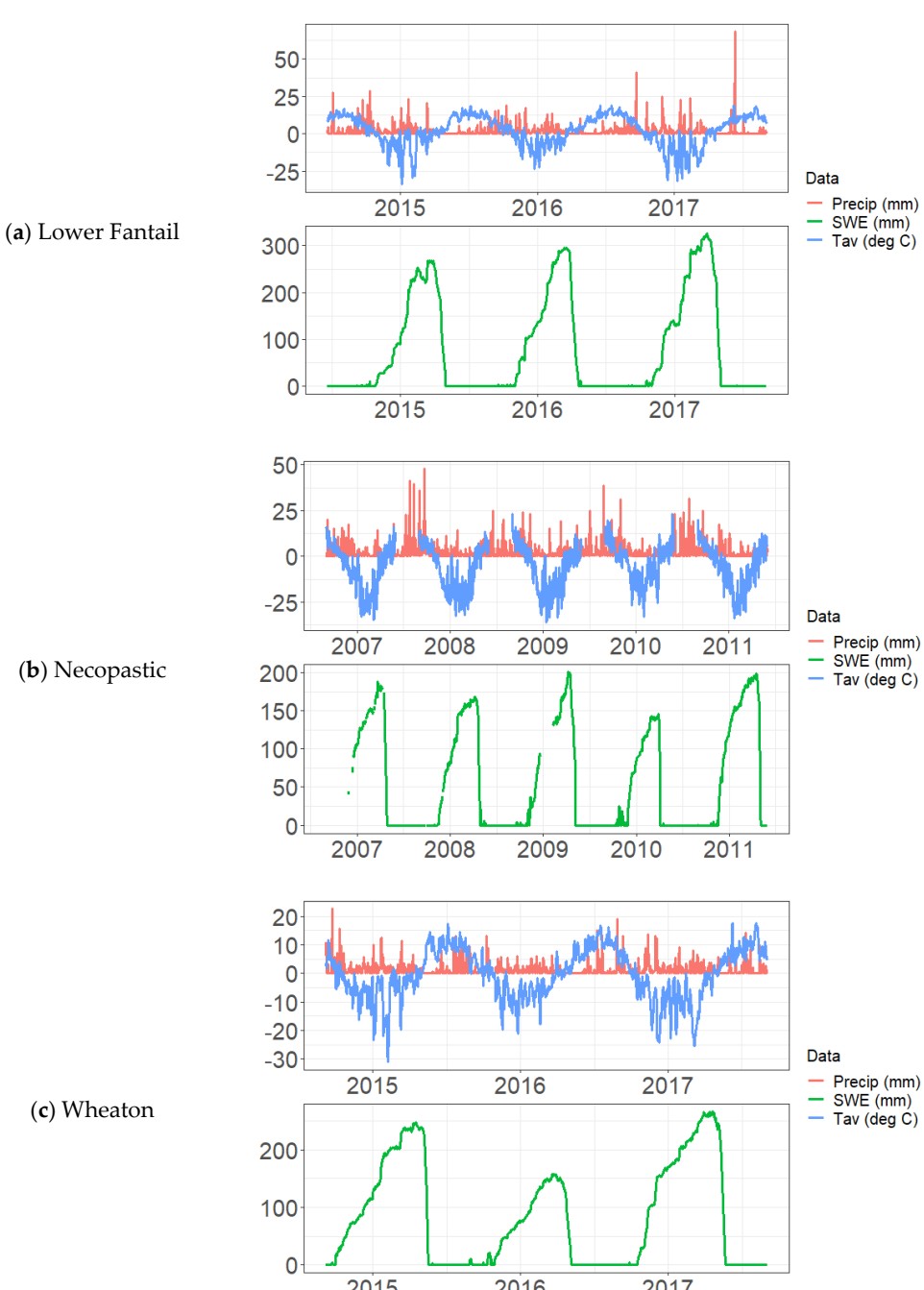

**Figure 3.** Daily precipitation, average air temperature, and *SWE* at the three stations.

**Table 3.** Cumulative precipitations and average temperature for each hydrological year.

| Station | Data | Y1 | Y2 | Y3 | Y4 | Y5 |
|---|---|---|---|---|---|---|
| Lower Fantail | Years | 2014/2015 | 2015/2016 | 2016/2017 | - | - |
| | Precipitation (mm) | 643 | 556 | 721 | - | - |
| | Average temperature (°C) | 1.8 | 1.5 | 2.0 | - | - |
| Necopastic | Years | 2006/2007 | 2007/2008 | 2008/2009 | 2009/2010 | 2010/2011 |
| | Precipitation (mm) | 803 | 855 | 840 | 819 | 462 |
| | Average temperature (°C) | 2.2 | 2.3 | 2.3 | 2.2 | 1.7 |
| Wheaton | Years | 2014/2015 | 2015/2016 | 2016/2017 | - | - |
| | Precipitation (mm) | 525 | 352 | 489 | - | - |
| | Average temperature (°C) | −0.2 | 0.6 | −1.9 | - | - |

## 3. Results

### 3.1. Sensitivity Analyses

The sensitivity analysis was conducted for the three stations. Figure 4 depicts the relative sensitivity index $IVAR_{i,50n}$ for each parameter for both models.

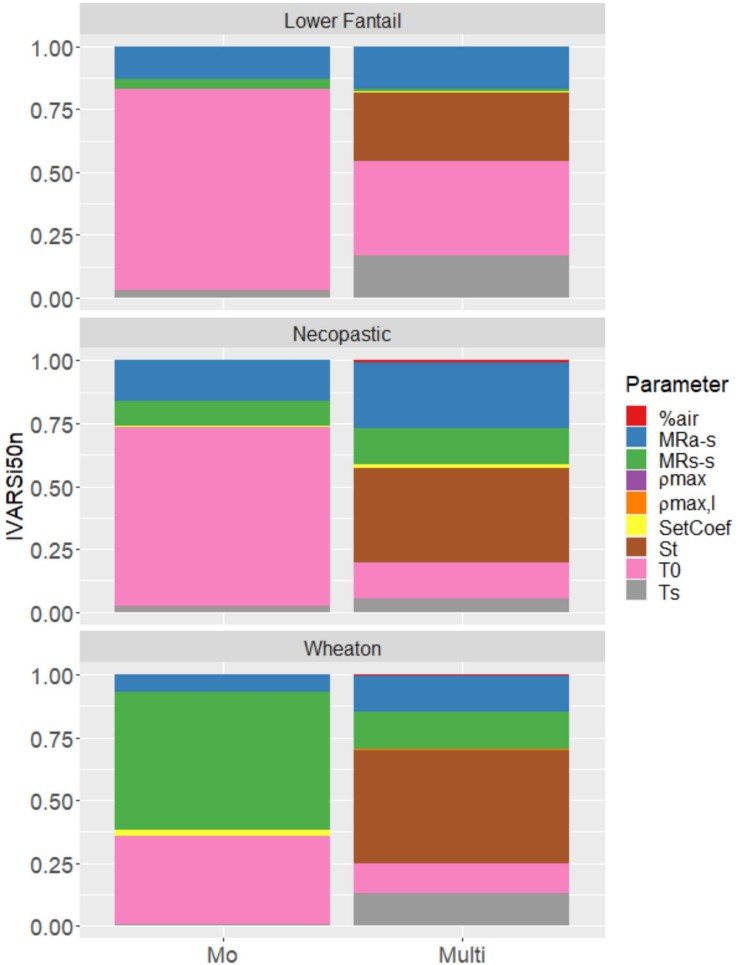

**Figure 4.** Normalized sensitivity analysis of the monolayer (Mo) and multilayer (Multi) snow model.

Before comparing parameter sensitivity differences between the two models, it is necessary to evaluate those of the three additional parameters of the Multi snow model. Notably, $S_t$, the new snow layer precipitation threshold, displayed high sensitivity and requires calibration, while $\rho_{max,l}$ and *%air* exhibited minimal sensitivity values and could therefore be set to a constant value prior to calibration. Both Dahe et al. [60] and Nishimura et al. [61] observed a maximum value of 550 kg.m$^{-3}$ for $\rho_{max,l}$. Considering its sensitivity and range of values set at 250–550 kg.m$^{-3}$ in the Mo model, it was set to 550 kg.m$^{-3}$ for the Multi model. As for *%air*, it was established as 10% of the snowpack depth in the Mo model. In the multilayer model MASiN [42], it was set at 8% of the volume of the snowpack not occupied by the *SWE* or the liquid water content, with some allowance possible for values varying between 5 and 10%. Würzer et al. [62] set a value of 3.5% of the snow depth in the SNOWPACK model. Taking these divergent values into account, *%air* was set at 10% of the snowpack height occupied by air in the Multi model.

The most sensitive phenomenon in the Mo model was located at the boundary between the atmosphere and snow. Two parameters, $T_0$ (the threshold temperature for considering melt due to radiation) and $MR_{a-s}$ (the degree-day rate of melt due to radiation), are crucial in this context. $Set_{Coef}$ (i.e., compression rate) and $T_s$ (threshold temperature for precipitation partitioning into rain and snow) are insensitive. By incorporating a

multilayer structure into the model, the significance of $T_s$ is given greater importance while simultaneously minimizing the relative sensitivity of the boundary phenomenon between the atmosphere and snow.

Figure 5 illustrates the daily relative sensitivity $IVAR_{i,50n}$ of both models at the three stations. The discontinuity arose from limited data over few years. The parameter factor space did not allow the Mo model to simulate snow during the summer season, in contrast to the Multi model.

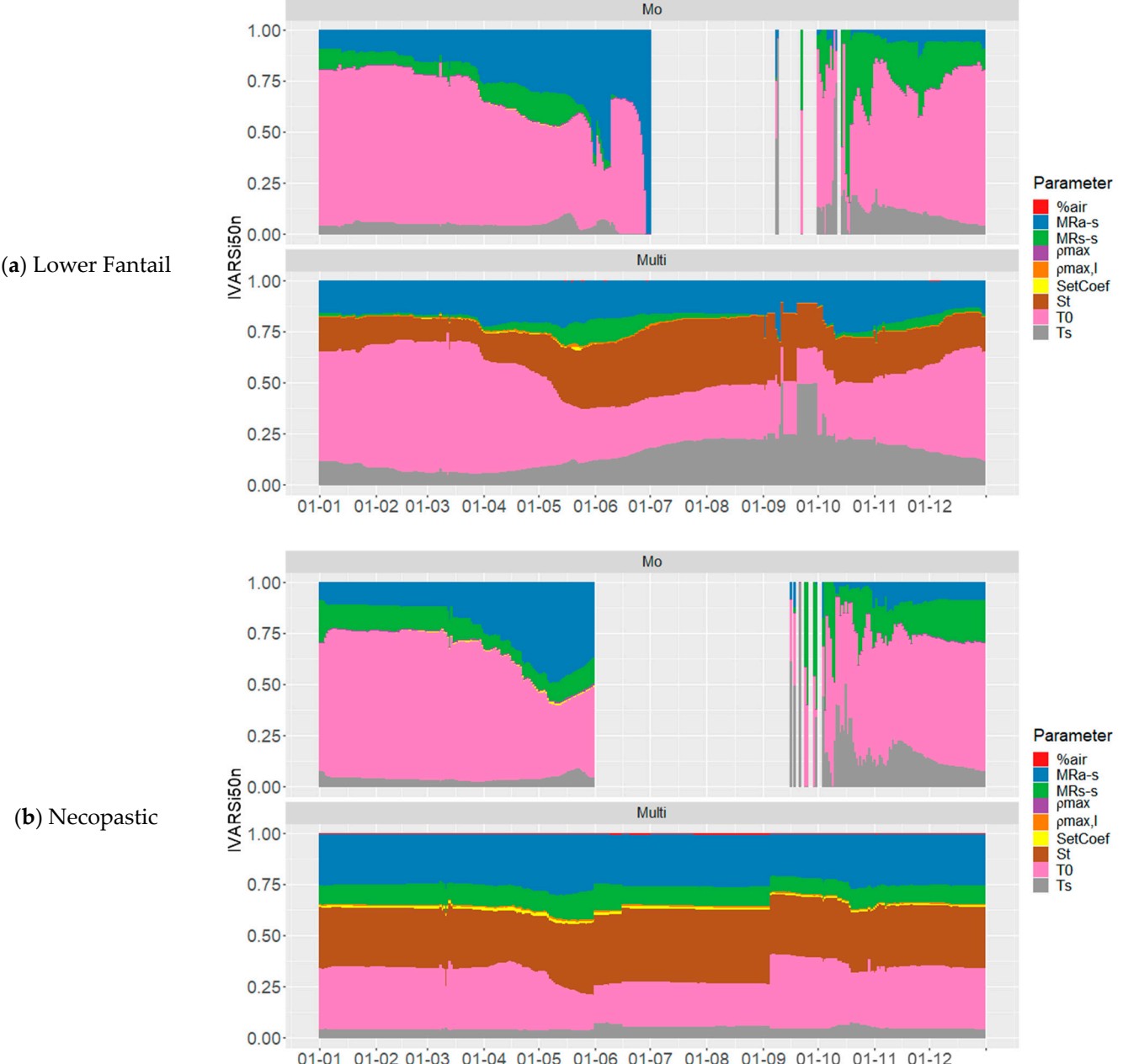

**Figure 5.** *Cont.*

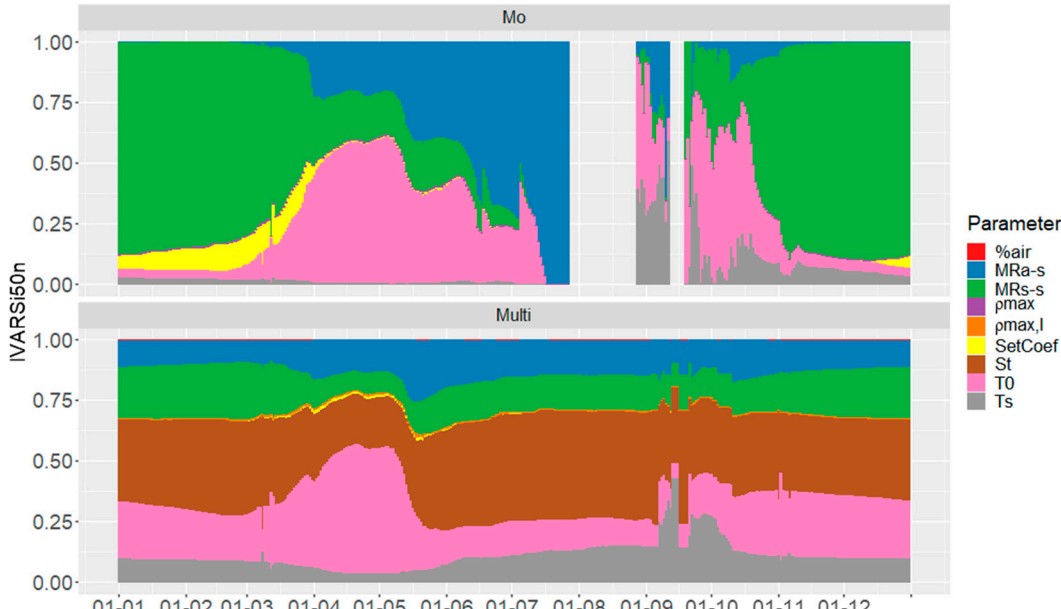

**(c)** Wheaton

**Figure 5.** Daily relative sensitivity analysis of the monolayer and multilayer models at the (**a**) Lower Fantail station, (**b**) Necopastic station, and (**c**) Wheaton station.

The seasonal phenomena are highlighted in both models. For the Mo model, the temperature threshold for separating precipitation ($T_s$) was quite sensitive early in the formation of the snowpack. The most significant phenomenon during spring was melting caused by radiation ($MR_{a-s}$). As the melting season drew to a close, melting from the soil became increasingly important ($MR_{s-s}$). The parameter $Set_{Coef}$ (settling rate) was also quite sensitive prior to the melting season, particularly at the Wheaton station.

For the Multi model, variations in sensitivity were less severe but still revealed the same seasonal phenomena as in the Mo model. However, the new snow layer precipitation threshold ($S_t$) served as a buffer during the melting period.

### 3.2. Modeling Performances—Validations

Figure 6 depicts the performances of the snow models for the top ten best parameter sets for the calibration and validation periods at the three GMON stations.

For the Lower Fantail station, the Wilcoxon test indicated no significant difference (*p*-value > 0.05) between the median of the models during the "Y23" combination calibration period, where the first year of data was used for validation, which was the driest and coldest year. For the remaining combinations, the Multi model improved median performances by 0.021 to 0.033 for the calibration period and by 0.125 to 0.223 for the validation period.

The performances of both models for the Necopastic station did not exhibit significant differences over the calibration period for combinations "Y1235" and "Y1245", and over the validation period for the combinations "Y1234" and "Y1245". However, during the calibration period, the Multi model boosted performance by 0.01 to 0.017 of KGE, and during the validation period, it improved by 0.009 to 0.154. The Mo model improved the performance by 0.012 for "Y2345" for the calibration period and by 0.012 for the validation period for "Y1345". Notably, there was no relationship with annual meteorological characteristics.

Conversely, for the Wheaton station, there was no significant difference between the models over the validation period for combination "Y12" or "Y23". However, the Mo model enhanced the performance by 0.03 for combination "Y13", which considered the driest and warmest year for validation. During the calibration period, the Mo model improved the performance by 0.008 to 0.04.

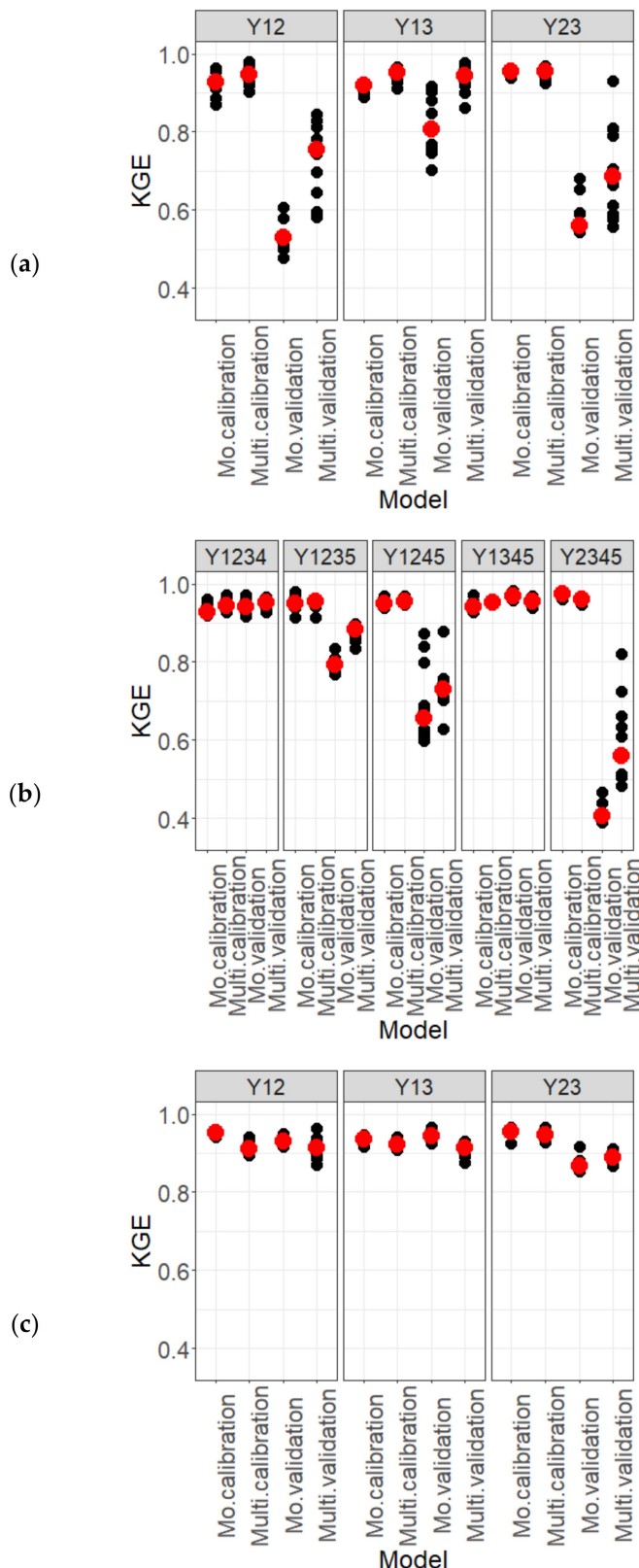

**Figure 6.** KGE values for Mo and Multi models for (**a**) Lower Fantail, (**b**) Necopastic, and (**c**) Wheaton. In red are the median performances of the top ten best parameter sets. The missing number in each column corresponds to the year used for validation; for example, Y12 means that year 3 was used for validation.

Of the eleven configurations detailed above for the calibration period, there were three cases where the snow models performed equally. It is important to note that this evaluation is objective and based solely on performance. The Mo model performed better in four configurations (with an average gain of 0.019 of KGE), whereas the Multi model performed better in the remaining four configurations (with an average gain of 0.020 of KGE). The gains were comparable across the calibration periods. Out of the configurations for the validation period, there were four cases where the snow models showed no difference. The Mo model performed better on two configurations (average gain: 0.021 of KGE), whereas the Multi model performed better on the remaining five configurations (average gain: 0.146 of KGE). The Multi model demonstrated a clear improvement in result consistency over the validation period.

*3.3. Modeling Performances—All Calibrations*

In the third part of this paper, calibration was performed using all years. For the Lower Fantail and Wheaton stations, the results of the Wilcoxon tests rejected the median equality hypothesis, yielding *p*-values of $9.8 \times 10^{-3}$ and $2 \times 10^{-3}$, respectively (with Multi median values of 0.95 and 0.92, respectively, and Mo model median values of 0.93 and 0.95, respectively). Conversely, for the Necopastic station, the medians (Multi: 0.95, and Mo: 0.96) are considered equal, given an $8.4 \times 10^{-2}$ *p*-value. Regarding the root mean squared error (RMSE) and the Nash–Sutcliffe efficiency (NSE), the Wilcoxon test failed to reject the null hypothesis that the medians are equal. Figure 7 illustrates the calibration performances (KGE, RMSE, and NSE values) of the top 10 sets of parameters obtained for each model at the three stations, as well as the coefficients derived from linear regression analysis.

It is evident that the slopes obtained from the Mo model had a narrower range than those of the Multi model during the snow accumulation (defined as the observed period between the first day of snow on the ground and the winter peak) and the melt period (defined as the observed period between the winter peak and the day when the snow cover has completely melted). Furthermore, the range increased more during the melting period compared to the accumulation period for each model.

Figure 8 depicts *SWE* simulations based on the top ten parameter sets for each model at the Lower Fantail station. Results for the two other stations can be found in Appendix F.

The results show minimal disparities in the optimal performances, with KGE values consistently exceeding 0.95 for the optimal sets of parameter values. Assessing robustness through the minimum values of the red interval indicated a similarity for both models. However, because of their inherent differences, *SWE* absolute values differed substantially between models. Notably, the Multi model showed more pronounced seasonal variability (red interval width), thereby enabling a more precise representation of the first winter peak at the Lower Fantail station with certain parameter sets, whereas the Mo model failed to represent adequately the observed *SWE* profiles.

Similarly, Figure 9 displays the range of snow height and density modeled by the top ten sets of parameter values for each model. Analyzing the snow height time series is relevant, as this variable is used for the *SWE* estimation in both models. The snow height series was overestimated by the Mo model, whereas the Multi model underestimated them, except for a few sets of parameters. This resulted in underestimated snow densities by the Mo model, as opposed to the output of the Multi model. It is evident that the Multi model overestimated the density during each phase of melting.

The modeled snow height and density time series for the Lower Fantail and Necopastic GMON stations are presented in Appendix F. Figure 10 shows the KGE values for snow height and density time series achieved by the top ten sets of parameter values, calibrated on *SWE* for both models.

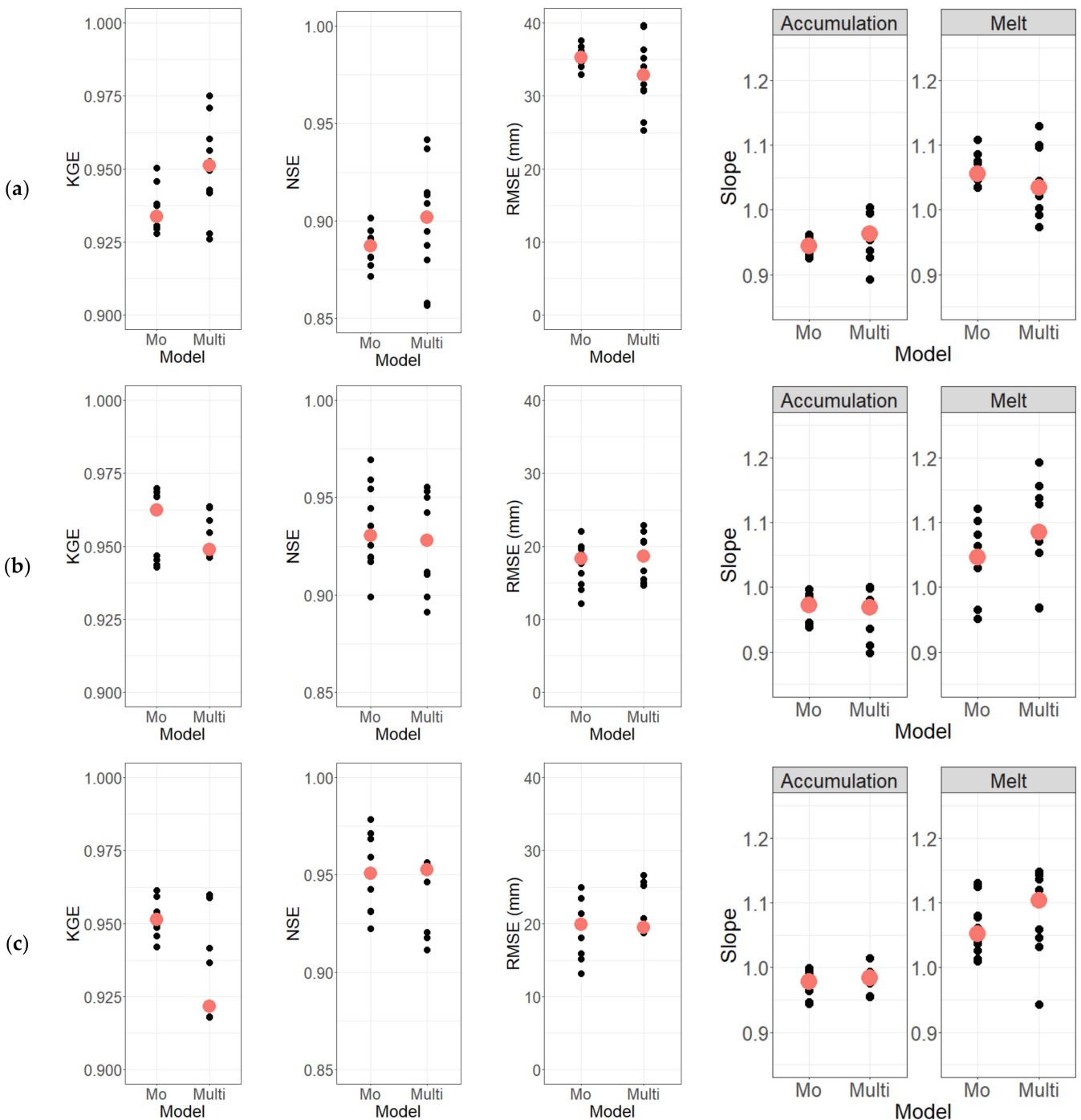

**Figure 7.** Modeling performances (KGE, RMSE, and NSE) and average rate of change (i.e., slope) of *SWE* during the snow accumulation and melt periods of the top ten sets of parameter values obtained for the multilayer snow model (Multi) and the monolayer model (Mo) for (**a**) the Lower Fantail station (LF), (**b**) the Necopastic station (Neco), and (**c**) the Wheaton GMON station (W). In orange is the median performance. KGE, RMSE, and NSE stand for Kling–Gupta efficiency, root mean squared error, and Nash–Sutcliffe efficiency, respectively.

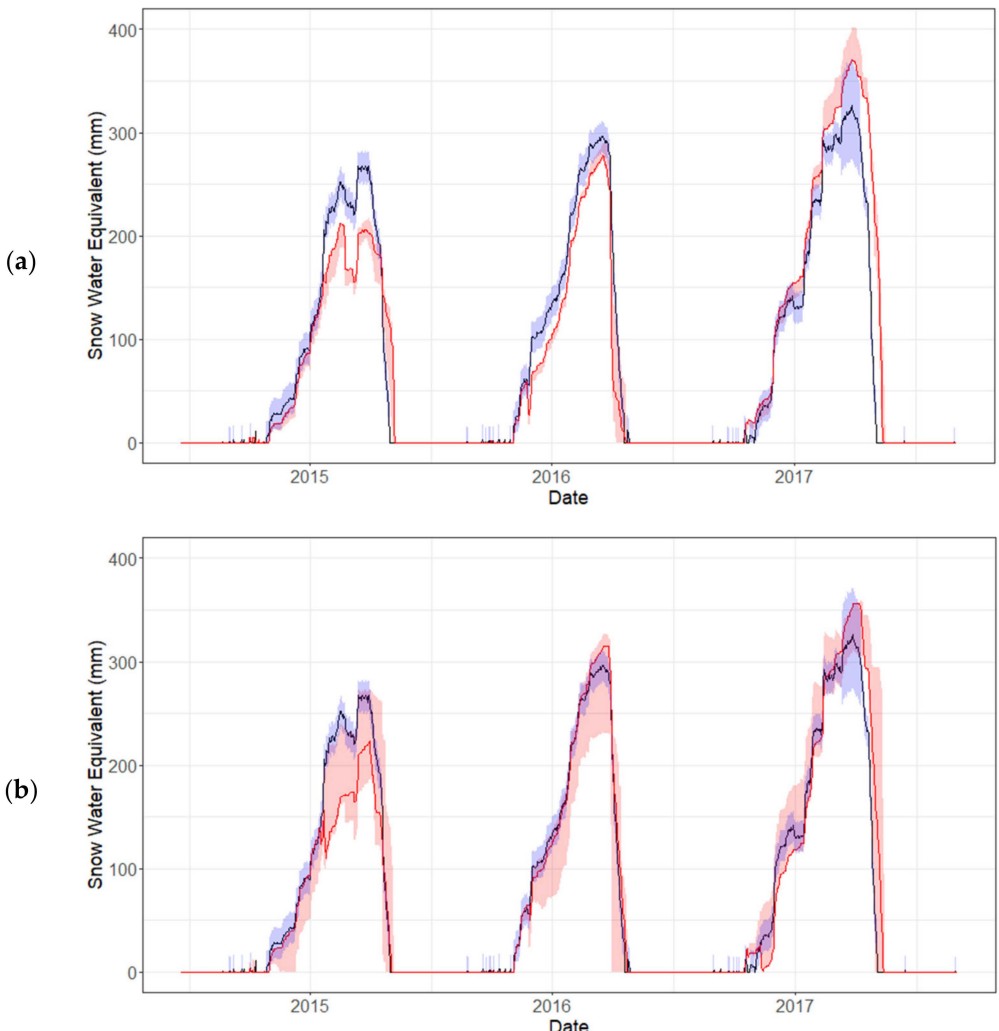

**Figure 8.** Modeled *SWE* time series at the Lower Fantail station for the (**a**) monolayer (Mo) and (**b**) multilayer (Multi) models. The red shaded interval shows the range of values provided by the top ten sets of parameters, with the red line for the best parameter set. The observed *SWE* time series is shown in black, while the blue interval depicts the measurement uncertainty.

Although the minimum performances can be considered unsatisfactory for each model, the median performances indicate that the Multi model more frequently generated physically accurate simulations (with a KGE around or greater than 0.5), whereas the acceptable results provided by the Mo model were achieved only by a few sets of parameters. Consequently, the Multi model can offer more parameter sets for *SWE*, providing satisfactory performances for snow height, compared to the Mo model. However, it is noteworthy that during the melting period, the densities of the snowpack layers remained high for the Multi model, incorporating layers of ice (density of 917 kg.m$^{-3}$) with thicknesses exceeding 20 cm.

Furthermore, the modeling of freezing rain was of little impact. Out of the ten best sets of parameter values obtained for each station, only one parameter set modeled this type of rain for Necopastic, and none for Lower Fantail and Wheaton. More importantly, during calibration, only 9.6% of the parameter sets accounted for any freezing rain event for the Necopastic GMON station, and none for the Lower Fantail and Wheaton stations.

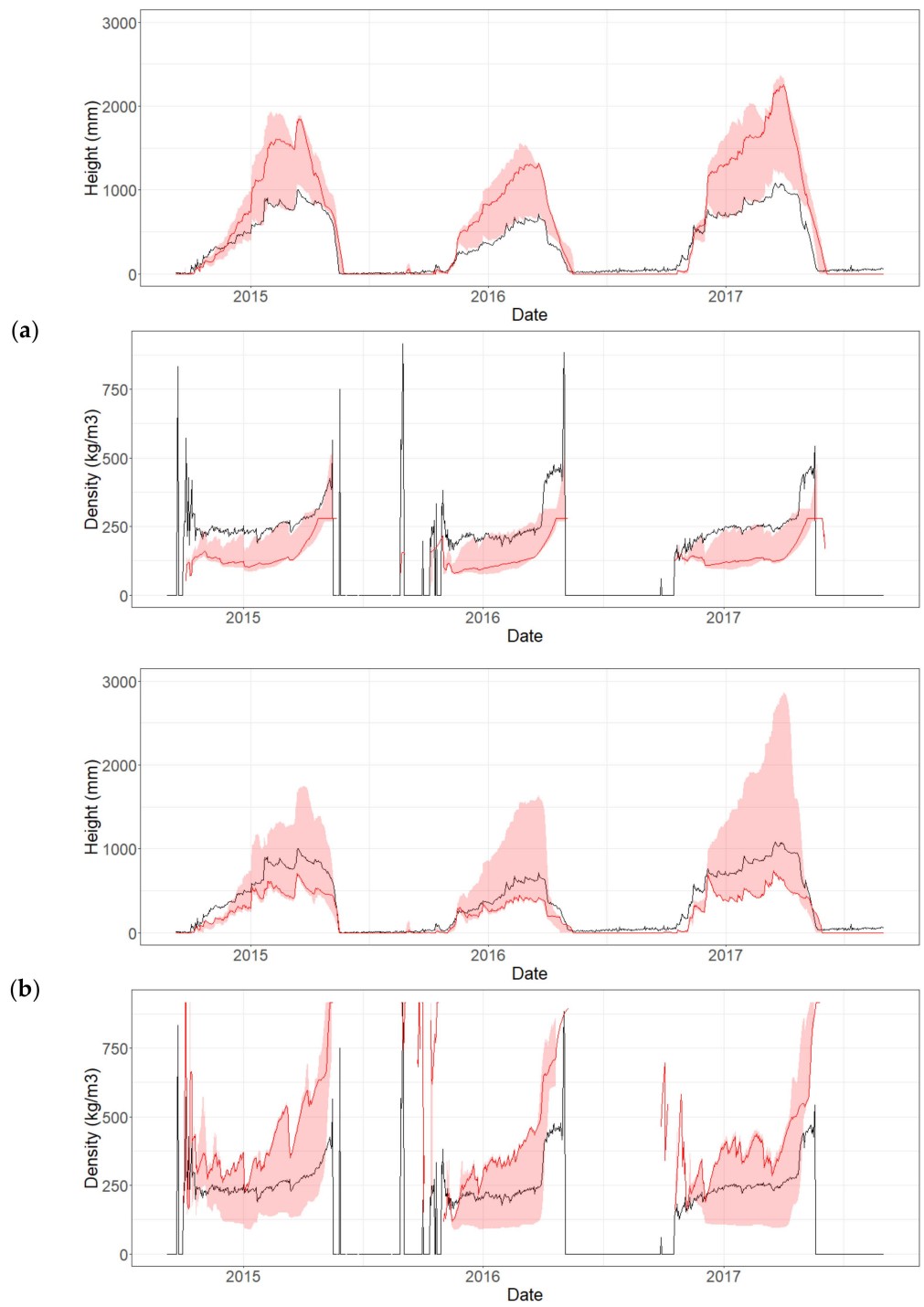

**Figure 9.** Modeled height and density time series at the Wheaton station for the (**a**) monolayer (Mo) and (**b**) multilayer (Multi) models. The red shaded interval shows the range of values provided by the top ten sets of parameters, with the red line for the best parameter set. The observed height and density time series are shown in black.

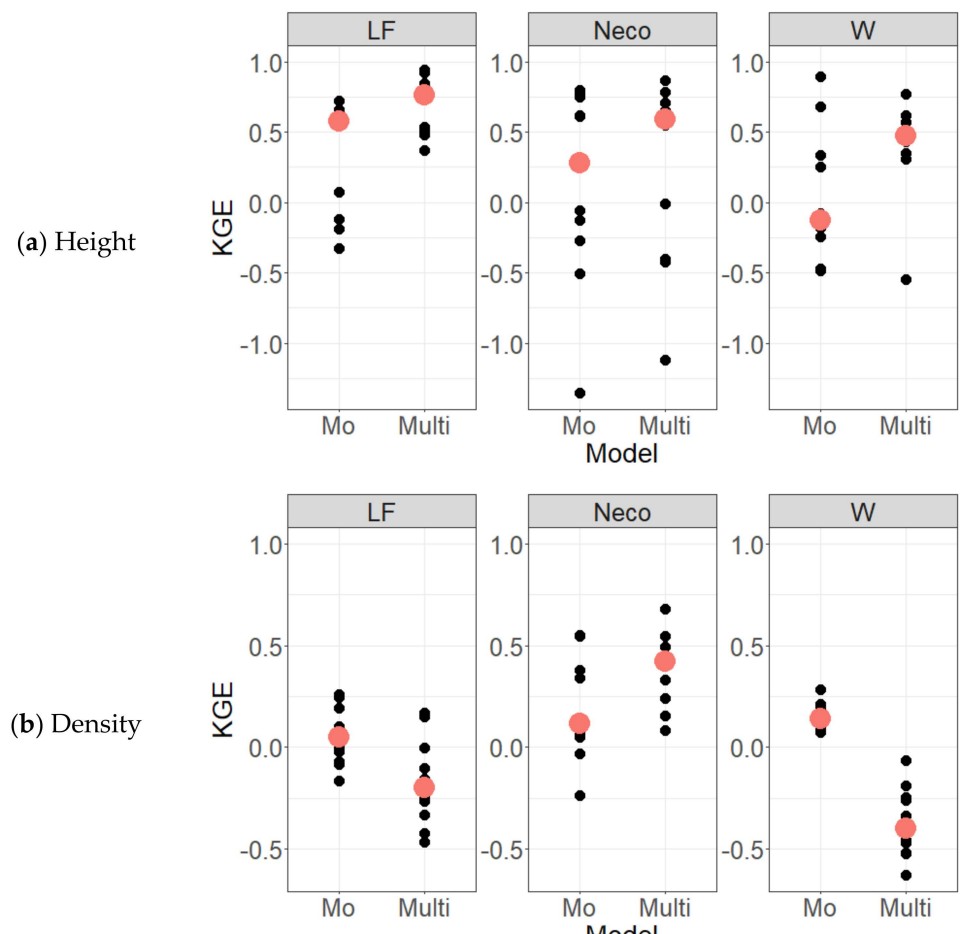

**Figure 10.** KGE values computed from the ten best parameter sets using time series of observed and modeled snow heights (**a**) and densities (**b**) for the Mo and Multi models. In orange is the median performance.

*3.4. Modeling Snowpack Characteristics*

Snowpack characteristics derived from the top ten sets of parameter values of each model were compared in terms of the onset and end dates of the snowpack, as well as the maximum *SWE* values and dates. The seasonal discrepancies between the modeled and observed data were analyzed across the top sets of parameter values in Table 4. This assessment provides insights into the equifinality of each feature of interest. For instance, the top 10 parameter sets presented here for each station and model yielded global KGE values greater than 0.9. However, snow peaks or melting periods may be modeled differently given the set of parameter values used.

**Table 4.** Medians of annual differences between observations and snowpack characteristics from the top ten best sets of parameter values of each model at the three GMON stations.

| Station | Lower Fantail | | Necopastic | | Wheaton | |
|---|---|---|---|---|---|---|
| **Models** | **Mo** | **Multi** | **Mo** | **Multi** | **Mo** | **Multi** |
| Onset date (days) | 3 | 4 | 3 | 3 | 3 | 4 |
| End date (days) | 8 | 7 | 4.5 | 4 | 6 | 1.5 |
| Maximum *SWE* date (days) | 5.5 | 7 | 9 | 11 | 1 | 8 |
| Maximum *SWE* relative difference (%) | 17 | 6.7 | 11 | 5.9 | 8.8 | 13.2 |

Both Mo and Multi onset dates showed consistent median deviations of 3–4 days from the observed data. The end date deviations were similar, except for the Wheaton station,

where the Multi model showed a 4-day improvement over the Mo model. Comparing with the Multi model, the maximum *SWE* dates were better represented with the Mo model by 1.5, 2, and 7 days for the Lower Fantail, Necopastic, and Wheaton stations, respectively. Notably, the Multi model outperformed the Mo model in representing the maximum *SWE*, particularly exhibiting a halved error at the Lower Fantail and Necopastic stations, but with a higher error at the Wheaton station.

As previously introduced, the Multi model uses a different approach to estimate snow albedo compared to the Mo model. Mo assumes that the albedo decays with time as a function of snowpack liquid water content, whereas Multi estimates albedo as a linear function based on the proportion of ice and air in the top snow layer. Figure 11 illustrates the albedo values of the top ten best parameter sets for both models for the Wheaton station (the albedos for the Lower Fantail and Necopastic stations are depicted in Appendix F). It can be observed that the estimated albedo for the Mo remained consistent across each parameter set, whereas more variations were observed for Multi. Although both approaches demonstrate a decreasing albedo over winter, Multi's behavior was consistent throughout the winter, except following a snowfall, which could have temporarily increased the albedo after the new snow blended in the uppermost layer or after adding a new layer. The decreasing albedo of Mo fluctuated within a certain range during winter until the spring melt, when it strongly decreased. Finally, the albedo of Multi was greater than that of Mo because it is calculated for the uppermost snow layer only, whereas Mo considers an equivalent albedo for the entire snow cover.

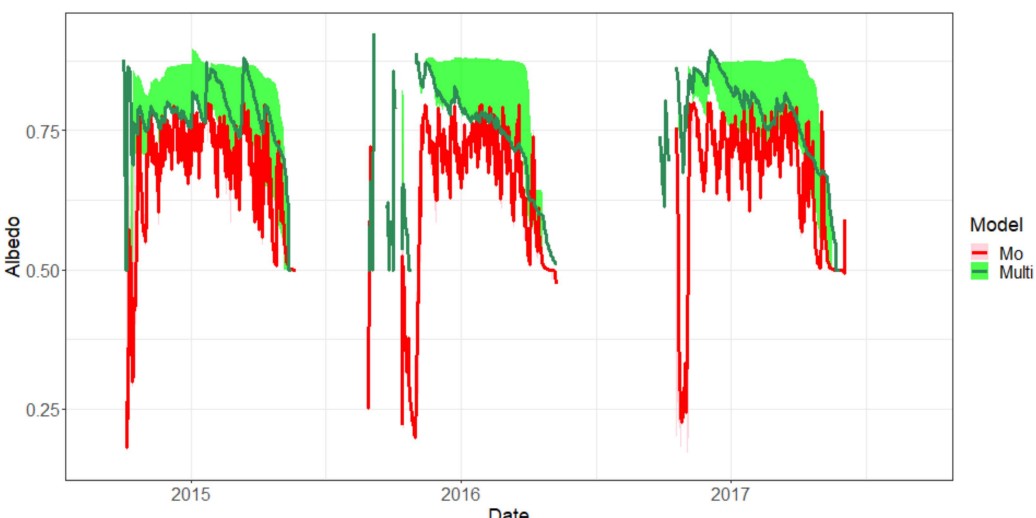

**Figure 11.** Albedo time series modeled by the top ten best sets of parameter values for the Mo model (pink envelope) and the Multi model (green envelope) for the Wheaton GMON station. The best parameter sets are depicted by the red and green lines for the Mo and Multi models, respectively.

## 4. Discussion

This paper has proposed a set of modifications to the monolayer snow model of HYDROTEL, including the integration of a multilayer structure, estimation of snowpack properties based on the proportion of ice and air, freezing rain modeling, and changes in compression and maximum water retention capacity. The modeling was assessed with respect to *SWE* modeling and other snow characteristics, such as snow height and density.

The sensitivity analysis indicated that amongst the changes implemented in the Mo model, the precipitation threshold for adding a snow layer ($S_t$) was highly sensitive, whereas the ratio of the volume of air that can be filled in by water (%*air*) and the settling maximum snow-layer density ($\rho_{max,l}$) were less sensitive. The addition of these parameters changed the hierarchy of sensitivity of the other parameters. For instance, whereas the melting temperature threshold at the air–snow interface became less sensitive ($T_o$), the melting

rate sensitivity at this interface ($MR_{a-s}$) increased. Similarly, the temperature precipitation threshold temperature ($T_s$) becoming more sensitive was deemed significant. In addition, the melting rate at the ground–snow interface ($MR_{s-s}$) became less sensitive, except for a slight increase in sensitivity for the Necopastic GMON station. These modifications rendered the Multi model more sensitive to phenomena at the snow–atmosphere interface. Furthermore, the melt rate at the snow–ground interface ($MR_{s-s}$) became generally less sensitive, emphasizing the influence of the atmosphere on snow melt rather than at the snow–ground interface. This change in behavior is consistent with observations made by Lackner et al. [63], who showed that temperature variations within the snowpack exhibit amplitudes more akin to those in the atmosphere than those at the ground level.

The calibration/validation strategies were based on 22 combinations when examining their respective periods separately. Among these, seven combinations showed no significant difference in performance between models. During the calibration period, increases in performance did not exceed 0.04. Thus, both models demonstrated similar levels of performance over this period. However, during the validation period, the Mo model's performance did not surpass 0.03, whereas the range of increased performance for the Multi model varied between 0.046 and 0.223. Notably, there was a subset of four combinations that exhibited an increase in performance of more than 0.1. Overall, the Multi snow model demonstrated greater robustness during the validation period compared to the Mo model. When both models were calibrated using the full datasets, with respect to their relative performances in reproducing *SWE*, the results highlighted some very good performances, with KGE values consistently greater than 0.9. Thus, neither model gained a clear advantage over the other. The reconstruction of precipitation records for the Lower Fantail and Wheaton stations may have contributed to these performances, providing the appropriate amount of water to the snowpack on the correct days until the melting period. However, for the Necopastic station, performances were still good even though precipitation records were not reconstructed. This suggests that the reconstruction of precipitation does not necessarily affect the conclusions of this paper. The modifications introduced in the Multi model made it possible to maintain a level of performance similar to that of the Mo model while also providing more flexibility for the computation of energy transfer within the snowpack, as suggested by the sensitivity of the additional parameter ($S_t$) on modeled *SWEs*. Furthermore, from a hydrological modeling perspective, snowpack melt rates are crucial for estimating streamflow, especially the maximum *SWE*, with snowpack heights being a somewhat secondary objective.

Although *SWE* modeling performances were comparable, model behaviors for snow heights were not. The Mo model tended to overestimate snow heights, whereas the Multi model tended to be consistent with observed heights or even slightly underestimate them. For the Mo model, the height is used solely to estimate compression while affecting thermal diffusivity; it can also be adjusted using the calibrated maximum density. In contrast, for the Multi model, although the height is used for compression, it is also used to compute snowpack density, which is required for computations of thermal diffusivity, albedo, and maximum water retention. Since energy transfer by radiation governs snowmelt, a low albedo increases this transfer. During spring snowmelt, minimizing snowpack height implies high densities—which is not surprising given that *SWE* is also equivalent to the product of snow height and relative snow density—which in return reduces albedo. An additional indication that simulated densities are larger than what may be observed in general can be inferred through a comparison reported by Keenan et al. [64] between simulated and observed density profiles using the SNOWPACK model. The densities they observed reached values of about 475 kg.m$^{-3}$ at ground level, whereas the Multi model formed snow layers limited to 550 kg.m$^{-3}$, or 917 kg.m$^{-3}$ for ice layers during the spring melt, with thicknesses exceeding 20 cm, which is unrealistic. Indeed, these densities are more akin to those observed for glaciers [65].

The attempt to model freezing rain indicated that this phenomenon seldom occurred for all the tested parameter sets. Indeed, the required condition that the atmospheric

temperature near the ground be negative may be too restrictive, and it is emphasized that atmospheric phenomena must be considered to model this type of precipitation as effectively as possible. However, it was decided to keep this phenomenon in the model, as it is a mean of creating snow layers under conditions of temperatures close to 0 °C.

The results of this study showed that transforming the Mo model into a Multi model improves the simulation of the end date of the snowpack as well as the seasonal maximum *SWE*, albeit at the expense of the occurrence date. Oreiller et al. [54] considered wind-induced snow transport as a plausible explanation for *SWE* discrepancies for the Necopastic station. This could also be a plausible hypothesis for discrepancies at the other GMON stations, but that remains to be validated. The different approaches used by the Multi and Mo models to estimate snow albedo can be interpreted in terms of the location where phenomena are assessed. For the Mo model, the albedo mimics the distribution of the radiative heat flux throughout the snow cover. In contrast, the approach used by the Multi model emphasizes the distribution of this flux throughout the top layer. Furthermore, the albedo of the Mo model varies within a certain range during winter before decreasing during the spring melt, whereas that of the Multi model decreases throughout the winter. Based on observations made by Gray et al. [66] and Stroeve et al. [67], the behavior of the Mo model albedo is more accurate, but the range of values of the Multi model remains coherent (albedo > 0.65 during winter). In other words, (i) the Mo albedo is for the whole snow cover; (ii) the observed albedo is based on upgoing and outgoing radiation measurements, which depends on the depth of snow penetrated by shortwave radiation; and (iii) the albedo of the Multi model is assumed to be that of the top snow layer only, regardless of the thickness.

## 5. Conclusions

The snow model of HYDROTEL is a daily monolayer (Mo) model combining degree-day and physics-based equations. This paper proposed a multilayer (Multi) alternative, modifying some of the fundamental equations while preserving the overall computational structure and limiting the addition of new calibration parameters. These modifications increased the sensitivity of processes occurring at the atmosphere–snow interface and the subsequent energy balance of each snow layer, improving the realism of the model. Although snow heights were overestimated by the Mo model, the Multi model more accurately depicted them, although some underestimation persisted. These underestimations resulted from the development of excessively dense, thick, and persistent snow layers during melting periods. Nonetheless, the vertical density profiles became consistent, with the densest layers located at ground level. Also, *SWE* modeling performances were very good (KGE consistently above 0.9) for both models, with the Multi snow model demonstrating more robustness during the validation period. By focusing on snowpack characteristics, the Multi model improved estimations of snowpack end dates and maximum *SWE* but compromised the modeled dates of the latter occurrence. These behavioral changes point towards the potential for improving snowmelt runoff and consequently spring peak flows, which are ultimately linked to the maximum *SWE*. As the frequency of the freezing rain events will, in all likelihood, increase in Eastern Canada given global warming [68], it would be relevant to find a parsimonious way to model these events. However, given that it is primarily an atmospheric phenomenon, the challenge remains. As the hydrological science community is becoming increasingly interested in rain-on-snow events [69–73], the suggested modifications can be viewed as a first step toward modeling them using the Multi version of the HYDROTEL snow model. From a structural standpoint, it may be beneficial to include a basal snow layer to emphasize the thermal discontinuity at ground level. Moreover, future work will involve integration of the multilayer snow model into HYDROTEL to evaluate the effect on stream flow modeling.

**Author Contributions:** Conceptualization, J.A. and A.N.R.; method, J.A., A.N.R. and E.F.; software, J.A.; validation, J.A., A.N.R. and E.F.; interpretation, J.A. and M.B.; data management, J.A.; visualization, J.A. and A.N.R.; drafting and edition, J.A., E.F., A.N.R. and M.B. All authors have read and agreed to the published version of the manuscript.

**Funding:** The authors wish to gratefully acknowledge the financial support of the Natural Sciences and Engineering Research Council of Canada (NSERC) and Yukon Energy (YE) through Collaborative (#CRDPJ 499954-16) and Applied (#CARD2 500263-16) Research and Development grants.

**Data Availability Statement:** Data available on request due to restrictions.

**Acknowledgments:** This project would not have been possible without substantial contributions from staff at the Yukon Research Center, namely, Brian Horton and Maciej Stetkiewicz, and at Yukon Energy, namely, Shannon Mallory, Kevin Maxwell, and Andrew Hall, as well as Mathieu Oreiller.

**Conflicts of Interest:** The authors declare no conflicts of interest.

## Appendix A. Snow Model Characteristics

**Table A1.** Snow model characteristics.

| Model | Design | Input Data | Considered Phenomena | Structure/Time Step | Reference |
|---|---|---|---|---|---|
| CEMANEIGE | Conceptual | - Atmospheric temperature<br>- Precipitation | - Accumulation<br>- Melt | Monolayer/ day | [74] |
| HBV | Physics-based Degree-day | - Atmospheric temperature<br>- Precipitation | - Accumulation<br>- Degree-day melt<br>- Latent heat flux | Monolayer/ day | [75–77] |
| SWAT | Physics-based Degree-day | - Atmospheric temperature (min and max)<br>- Precipitation | - Accumulation<br>- Degree-day melt<br>- Sublimation | Monolayer/ day | [78,79] |
| HYDROTEL | Physics-based Degree-day | - Atmospheric temperature (min and max)<br>- Precipitation | - Accumulation<br>- Compression<br>- Mixed (radiation and degree-day melt)<br>- Precipitation heat<br>- Soil heat<br>- Sensible heat flux<br>- Water retention | Monolayer/ day | [28] |
| VIC | Physics-based Complex | - Atmospheric temperature<br>- Precipitation<br>- Relative humidity (may be estimated)<br>- Short- to long-wave radiations (may be estimated)<br>- Wind speed | - Accumulation<br>- Compression<br>- Precipitation heat<br>- Turbulent heat flux (Sensible and latent)<br>- Radiation<br>- Water retention | Bilayer/ hourly to daily | [80] |
| CROCUS | Physics- based complex | - Atmospheric temperature<br>- Precipitation<br>- Relative humidity<br>- Short- and long-wave radiations<br>- Wind speed | - Accumulation<br>- Compression<br>- Heat conduction<br>- Metamorphism<br>- Precipitation heat<br>- Radiations<br>- Runoff and intra-snow cooling<br>- Soil heat<br>- Sublimation<br>- Turbulent heat flux (sensible and latent)<br>- Wind transport | Multilayer/ hour | [81,82] |

**Table A1.** *Cont.*

| Model | Design | Input Data | Considered Phenomena | Structure/Time Step | Reference |
|-------|--------|-----------|---------------------|--------------------|-----------|
| MASiN | Physics-based Complex | - Atmospheric temperature<br>- Precipitation<br>- Relative humidity<br>- Wind speed | - Accumulation<br>- Soil heat<br>- Cloud cover<br>- Compression<br>- Conduction<br>- Radiation (estimation)<br>- Turbulent heat flux (sensible and latent)<br>- Water retention | Multilayer/hour | [42] |
| SnowPack | Physics-based Complex | - Atmospheric temperature<br>- Precipitation<br>- Relative humidity<br>- Wind speed and direction | - Accumulation<br>- Compression<br>- Conduction<br>- Turbulent heat flux (sensible and latent)<br>- Radiation (estimation)<br>- Water retention | Multilayer/10 min to day | [83,84] |
| SNOWPACK | Physics-based Complex | - Atmospheric temperature<br>- Precipitation<br>- Relative humidity<br>- Short- and long-wave radiations<br>- Wind speed | - Accumulation<br>- Compression<br>- Microstructure<br>- Precipitation heat<br>- Radiation<br>- Runoff<br>- Subsurface melt<br>- Surface haze<br>- Surface melt<br>- Turbulent heat flux (sensible and latent)<br>- Wind erosion<br>- Wind transport | Multilayer/hour | [31,85,86] |

**Appendix B. Energy Balance Terms of the HYDROTEL Monolayer Snow Model**

The different terms of the energy balance equation of HYDROTEL's monolayer snow model are described below.

The heat input from rain, $u_r$, is computed as follows:

$$u_r = \rho_w \left( C_f + C_w \frac{T_{max} + T_{min}}{2} \right) R \tag{A1}$$

where $\rho_w$ is the density of water (1000 kg.m$^{-3}$); $C_f$ is the latent heat of fusion of water (335,000 J.kg$^{-1}$); $C_w$ is the specific heat capacity of water (4184 J.kg$^{-1}$.°C$^{-1}$); $T_{min}$ and $T_{max}$ are the minimum and maximum air temperatures (°C), respectively; and $R$ is the daily rainfall rate (m.s$^{-1}$).

The heat input from the ground, $u_{s-s}$, is computed as follows:

$$u_{s-s} = \rho_w C_f \frac{MR_{s-s}}{86400} \tag{A2}$$

where $MR_{s-s}$ is the melting rate at the snow–ground interface (m.day$^{-1}$), and 86,400 is the conversion from day to seconds.

The snow heat deficit, $u_s$, is computed as follows:

$$u_s = \rho_w C_s \frac{T_{max} + T_{min}}{2} S \tag{A3}$$

where $C_s$ is the specific heat capacity of snow (2093.4 J.kg$^{-1}$.°C$^{-1}$), and $S$ is the daily snowfall rate (m.s$^{-1}$).

Heat loss by conduction and heat gain by radiation are enabled depending on the temperature threshold for radiation heat gain $T_0$. Indeed, if the daily average air temperature is lower than $T_0$, the conduction heat losses are estimated; otherwise, the heat gain estimation by radiation is enabled. Heat loss by conduction is estimated using the solution for heat transfer in a semi-infinite material with air temperature as a Dirichlet boundary condition. Thermal diffusivity is computed using estimations of the conductivity and depth of snow. The heat deficit is then updated using the snow temperature resulting from the conductive heat loss.

The radiation heat input, $u_{a-s}$, is computed as follows:

$$u_{a-s} = \rho_w C_f \frac{M_{pot}}{86400} \tag{A4}$$

where $M_{pot}$ is the potential melting rate due to radiation (m.day$^{-1}$), computed as follows:

$$M_{pot} = I \, MR_{a-s} \left( \frac{T_{max} + T_{min}}{2} - T_0 \right)(1 - \alpha) \tag{A5}$$

where $I$ is a radiation index, $MR_{a-s}$ is the melting rate at the air–snow interface (m.day$^{-1}$.$^\circ$C$^{-1}$), and $\alpha$ is the snow albedo.

The radiation index is the ratio of the index for a sloped surface to that of a flat surface [87]. The snow albedo is computed using the snowpack and fresh snowfall albedos, accounting for the exponential decay of radiation penetration within the snowpack [28]. The equations are presented in Appendices B and C.

When the snowpack melts, water is retained within the medium and is considered frozen at the next computational time step. The phase change then warms up the snowpack as follows:

$$u_{ac} = \rho_w C_f \frac{AR}{86400} \tag{A6}$$

where $AR$ is the water retained within the snowpack of the previous day (m.day$^{-1}$). It is computed using Equation (7) from the maximum water retention capacity estimated in Equation (6).

**Appendix C. Radiation Index Equations of the Monolayer Snow Model**

$\theta$ is the GMON station latitude in radians:

$$\theta = \frac{lat}{rad1} \tag{A7}$$

where $lat$ is the GMON station latitude ($^\circ$), and $rad1$ is the conversion factor from radians to degrees ($\approx 57.295779513^\circ$.rad$^{-1}$ = $(180^\circ)/\pi$.rad$^{-1}$).

$k$ is the slope angle (rad):

$$k = arctan(slope) \tag{A8}$$

where slope is the ground slope (rad).

$h$ is the surface azimuth angle (rad):

$$h = \frac{(495 - 45ori)360}{rad1} \tag{A9}$$

where $ori$ is the ground orientation (1 for east, 2 for north/east, 3 for north, . . ., and 8 for south/east). Detailed information is available in Rousseau et al. [88].

$\theta_1$ is the equivalent slope latitude (rad):

$$\theta_1 = arcsin(sin(k) \, cos(h) \, cos(\theta) + cos(k) \, sin(\theta)) \tag{A10}$$

$\alpha$ is the longitude variation between the slope and its horizontal surface:

$$\alpha = arctan\left(\frac{sin(k)\ sin(h)}{cos(k)\ cos(\theta) - cos(h)\ sin(k)\ sin(\theta)}\right) \tag{A11}$$

$e2$ is the Sun's/Earth's distance to its average on a specific day:

$$e2 = \left(1 - exc\ cos\left(\frac{day - 4}{deg1}\right)\right)^2 \tag{A12}$$

where $exc$ is the Earth's orbit eccentricity (=0.01673), $day$ is the Julian day, and $deg1$ ($\approx$58.1313429644 day.rad$^{-1}$ = $2\pi/365.25$), 4 January, is the Earth at its perihelion.

$i_{e2}$ is the solar constant as a function of the Earth–Sun distance (W.m$^{-2}$):

$$i_{e2} = \frac{i0}{e2} \tag{A13}$$

where $i0$ is the solar constant (=1361 W.m$^{-2}$).

$decli$ is the solar declination (rad), which is the angle between solar rays and the plane of the equator:

$$decli = 0.410152374218 sin\left(\frac{day - 80.25}{deg1}\right) \tag{A14}$$

$tampon$ and $tampon1$ are the angles (rad) that correspond to the sunshine duration on a flat surface and on a sloped surface, respectively:

$$tampon = -tan(\theta)\ tan(decli) \tag{A15}$$

$$tampon1 = -tan(\theta_1)\ tan(decli) \tag{A16}$$

$dur_{hor}$ is the sunshine duration on a flat surface:

$$dur_{hor} = 0\ if\ tampom > 1 \tag{A17a}$$

$$dur_{hor} = 12\ if\ tampon < -1 \tag{A17b}$$

$$dur_{hor} = \frac{arccos(tampon)}{w}\text{otherwise} \tag{A17c}$$

where $w$ is the Earth's angular speed (15$^\circ$.h$^{-1}$ =15/rad1 rad.h$^{-1}$).

$dur_{slp}$ is the sunshine duration on a sloped surface:

$$dur_{slp} = 0\ if\ tampon1 > 1 \tag{A18a}$$

$$dur_{slp} = 0\ if\ tampon1 < -1 \tag{A18b}$$

$$dur_{slp} = \frac{arccos(tampon1)}{w}\text{otherwise} \tag{A18c}$$

$t1_{slp}$ and $t2_{slp}$ are the irradiation starting and end times on a sloped ground, respectively.

$$t1_{slp} = -dur_{slp} - \frac{\alpha}{w} \tag{A19a}$$

$$t1_{slp} = -dur_{hor}\ if\ t1_{pte} < -dur_{hor} \tag{A19b}$$

$$t2_{slp} = dur_{slp} - \frac{\alpha}{w} \tag{A20a}$$

$$t2_{slp} = dur_{hor}\ if\ t2_{slp} > dur_{hor} \tag{A20b}$$

$t1_{hor}$ and $t2_{hor}$ are the irradiation starting and end times on flat ground, respectively.

$$t1_{hor} = -dur_{hor} \tag{A21a}$$

$$t2_{hor} = dur_{hor} \tag{A21b}$$

$i_{j1}$ and $i_{j2}$ are the radiation for a flat and a sloped surface, respectively.

$$i_{j1} = 0 \text{ if } t1_{hor} > t2_{hor} \tag{A22a}$$

$$i_{j1} = 3600 \, i_{e2} \left( (t2_{hor} - t1_{hor})sin(\theta)sin(decli) + \frac{cos(\theta)cos(decli)(sin(w \, t2_{hor}) - sin(w \, t1_{hor}))}{w} \right) \text{ otherwise} \tag{A22b}$$

$$i_{j2} = 0 \text{ if } t1_{sip} > t2_{sip} \tag{A23a}$$

$$i_{j2} = 3600 \, i_{e2} \left( \left(t2_{slp} - t1_{slp}\right)sin(\theta_1)sin(decli) + \frac{cos(\theta_1)cos(decli)\left(sin\left(w \, t2_{slp} + \alpha\right) - sin\left(w \, t1_{slp} + \alpha\right)\right)}{w} \right) \text{ otherwise} \tag{A23b}$$

$I$ is the radiation index.

$$I = \left| \frac{ij_2}{i_{j1}} \right| \text{ if } i_{j1} \neq 0 \tag{A24a}$$

$$I = 1 \text{ otherwise} \tag{A24b}$$

### Appendix D. Albedo Equations of the Monolayer Snow Model

*wet* stands for a wet snowpack:

$$wet = 1 \text{ if } R > 0 \text{ or } T > 0 \tag{A25a}$$

$$wet = 0 \text{ otherwise} \tag{A25b}$$

where $R$ is rainfall, and $T$ is the snow temperature (relative to the heat deficit).

With snow on the ground:

A maximum snowpack albedo $alb_{t+1}$ is computed relative to the snowfall's or snowpack's state of humidity.

$$alb_{t+1} = (1 - exp(-0.5 \, eq_{snow}))0.8 + (1 - (1 - exp(-0.5 \, eq_{snow}))) \left( 0.5 + (alb - 0.5)exp \left( -0.2 \frac{pdth}{24}(1 + wet) \right) \right) \tag{A26}$$

where $eq_{snow}$ is the snowfall water equivalent (mm), $alb$ is the snowpack albedo of the previous time step, and $pdth$ is the time step's number of hours.

*beta*2 is the snowpack radiation penetration exponential decay coefficient.

$$beta2 = 0.2 \text{ if } alb < 0.5 \tag{A27a}$$

$$beta2 = 0.2 + (alb - 0.5) \text{otherwise} \tag{A27b}$$

$$alb = (1 - exp(-beta2 \, st_{snow}))alb_{t+1} + (1 - (1 - exp(-beta2 \, st_{snow})))0.15 \tag{A28}$$

where $st_{snow}$ is the snowpack water equivalent (mm).

Without snow on the ground:

$$alb = (1 - exp(-0.5 \, eq_{snow}))0.8 + (1 - (1 - exp(-0.5 \, eq_{snow})))0.15 \tag{A29}$$

### Appendix E. Relationships between the Densities of Snow, Ice, and Air

The mass of a composite material is that of its constituent elements. The mass of snow as a mixture of ice and air is computed as follows:

$$W_s = W_i + W_a \tag{A30}$$

where $W_s$, $W_i$, and $W_a$ are the snow, ice, and air weights (kg), respectively.

The snow density is estimated for a snow volume that is the sum of the ice and air volumes.

$$\rho_s = \frac{W_s}{V_i + V_a} = \frac{W_i}{V_i + V_a} + \frac{W_a}{V_i + V_a} \tag{A31}$$

where $\rho_s$ is the snow density (kg.m$^{-3}$), and $V_i$ and $V_a$ are the ice and air volumes (m$^3$), respectively.

Per the definition of density, $W_i = V_i \rho_i$ and $W_a = V_a \rho_a$:

$$\rho_s = \frac{V_i}{V_i + V_a}\rho_i + \frac{V_a}{V_i + V_a}\rho_a \tag{A32}$$

where $\rho_i$ and $\rho_a$ are the ice and air densities (kg.m$^{-3}$), respectively.

This equation then shows that by considering snow a composite material, its density can be related to the densities of ice and air, with coefficients corresponding to the respective proportions. In general, this amounts to considering that there is the following relationship:

$$\rho_s = A\rho_i + B\rho_a \ (avec \ A + B = 1) \tag{A33}$$

Since the volumes of ice and air are not explicitly estimated in the snow models proposed in this paper, and knowledge of the volumetric proportions A and B is necessary, an alternative method must be used:

$$\rho_s = A\rho_i + (1 - A)\rho_a \tag{A34}$$

Thus, the volumetric proportion of ice $A$ in the snow can be estimated from equation $A + B = 1$ as follows:

$$A = \frac{\rho_s - \rho_a}{\rho_i - \rho_a} \tag{A35}$$

Thus, the volumetric proportion of air $B$ in the snow can be estimated from equation $A + B = 1$; that is:

$$B = \frac{\rho_i - \rho_s}{\rho_i - \rho_a} \tag{A36}$$

Thus, the knowledge or estimation of the densities of ice, air, and snow enables the derivation of the volumetric proportions of ice and air within the snow from Equations (A35) and (A36), respectively.

**Appendix F. Results**

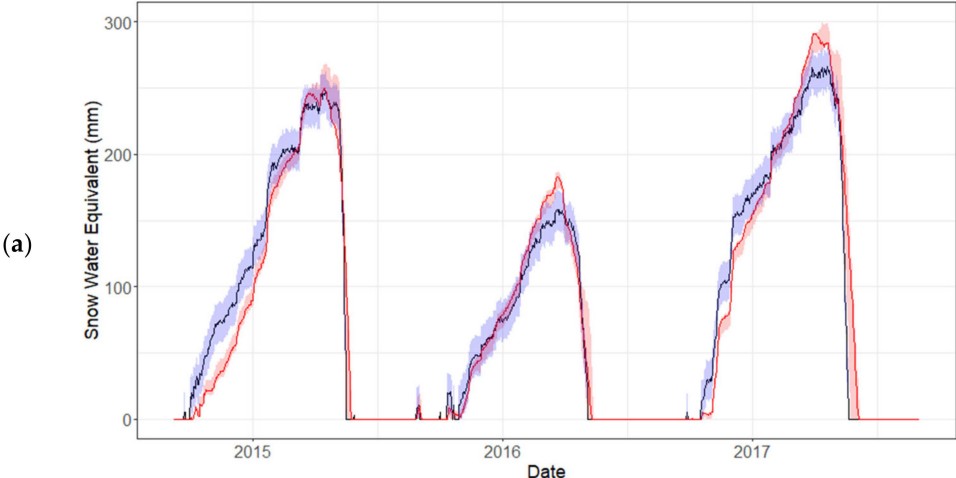

**Figure A1.** *Cont.*

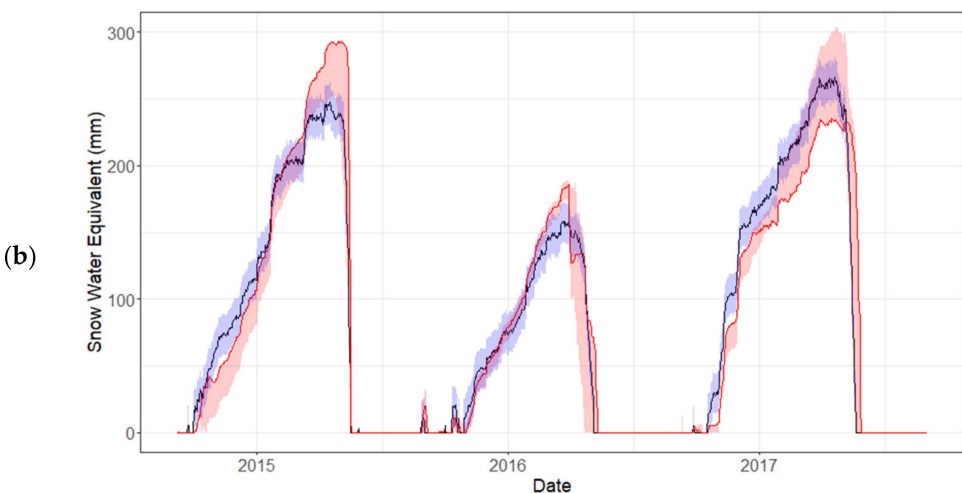

**Figure A1.** Modeled *SWE* series at the Wheaton station (W) for the (**a**) monolayer (Mo) and (**b**) multilayer (Multi) models. The red shaded interval shows the range of values provided by the top ten sets of parameters values. The observed *SWE* time series is shown in black, while the blue interval depicts the measurement uncertainty.

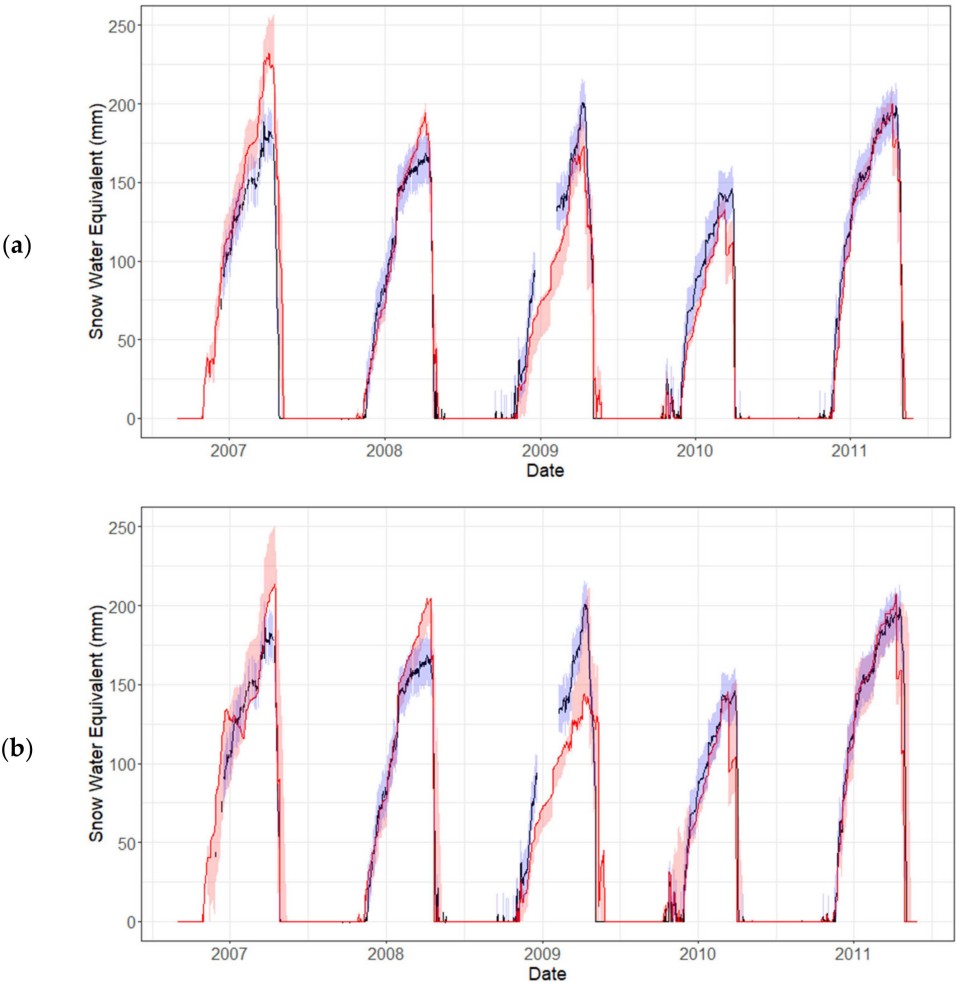

**Figure A2.** Modeled *SWE* series at the Necopastic station for the (**a**) monolayer (Mo) and (**b**) multilayer (Multi) models. The red shaded interval shows the range of values provided by the top ten sets of parameters values. The observed *SWE* time series is shown in black, while the blue interval depicts the measurement uncertainty.

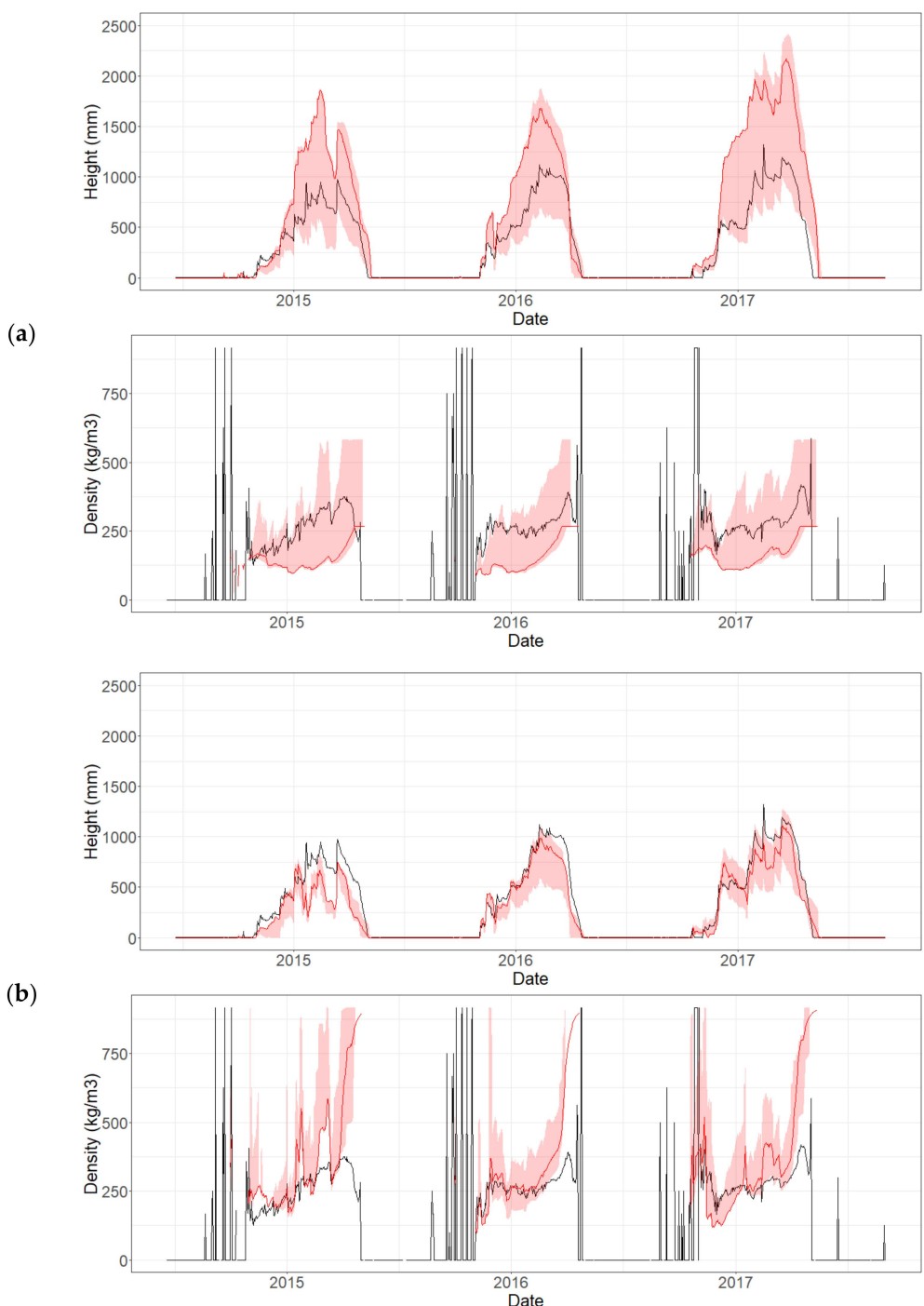

**Figure A3.** Modeled height and density series at the Lower Fantail station (LF) for the (**a**) monolayer (Mo) and (**b**) multilayer (Multi) models. The red shaded interval shows the range of values provided by the top ten sets of parameters values. The observed height and density time series is shown in black.

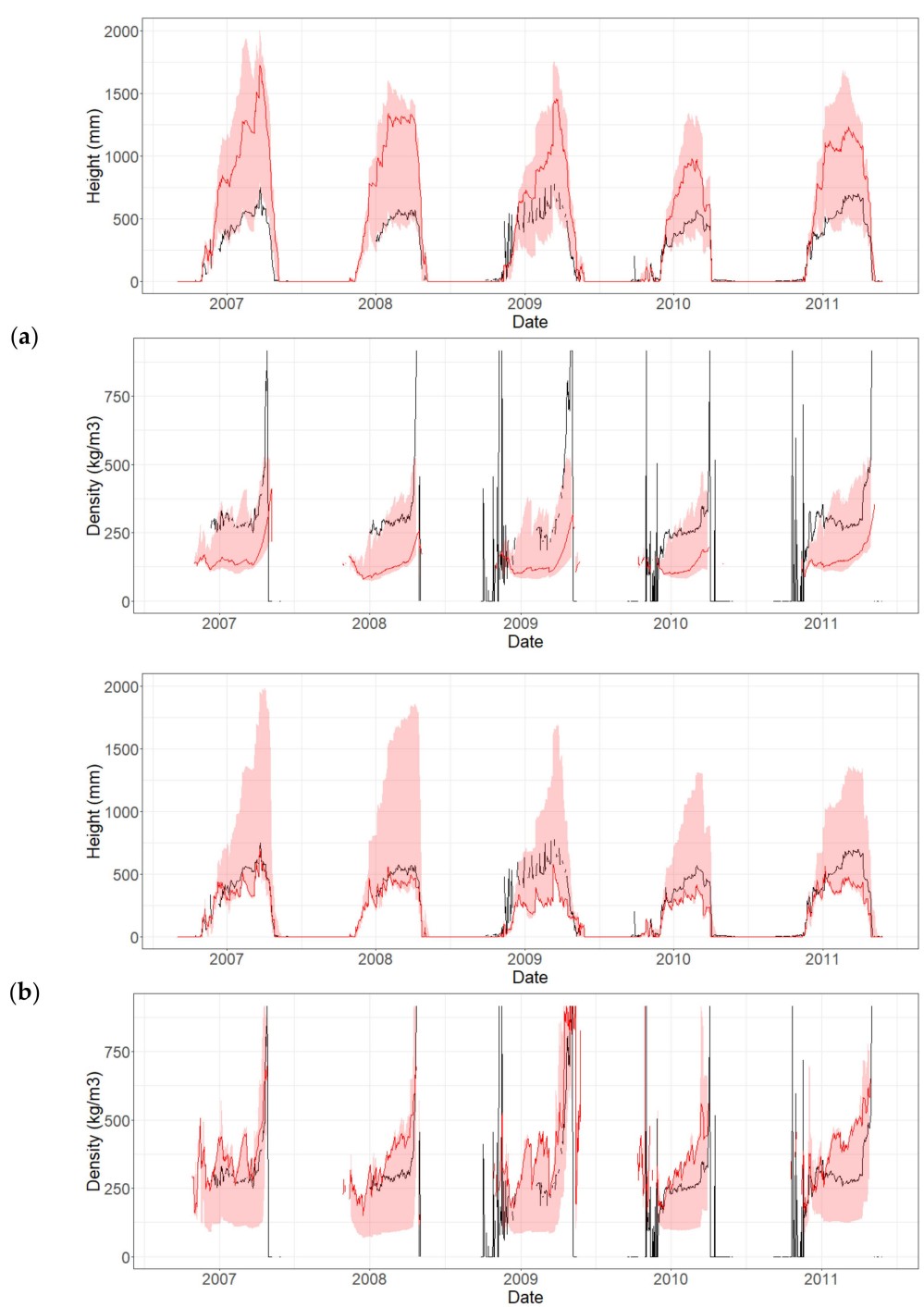

**Figure A4.** Modeled height and density series at the Necopastic station for the (**a**) monolayer (Mo) and (**b**) multilayer (Multi) models. The red shaded interval shows the range of values provided by the top ten sets of parameters values. The observed height time and density series is shown in black.

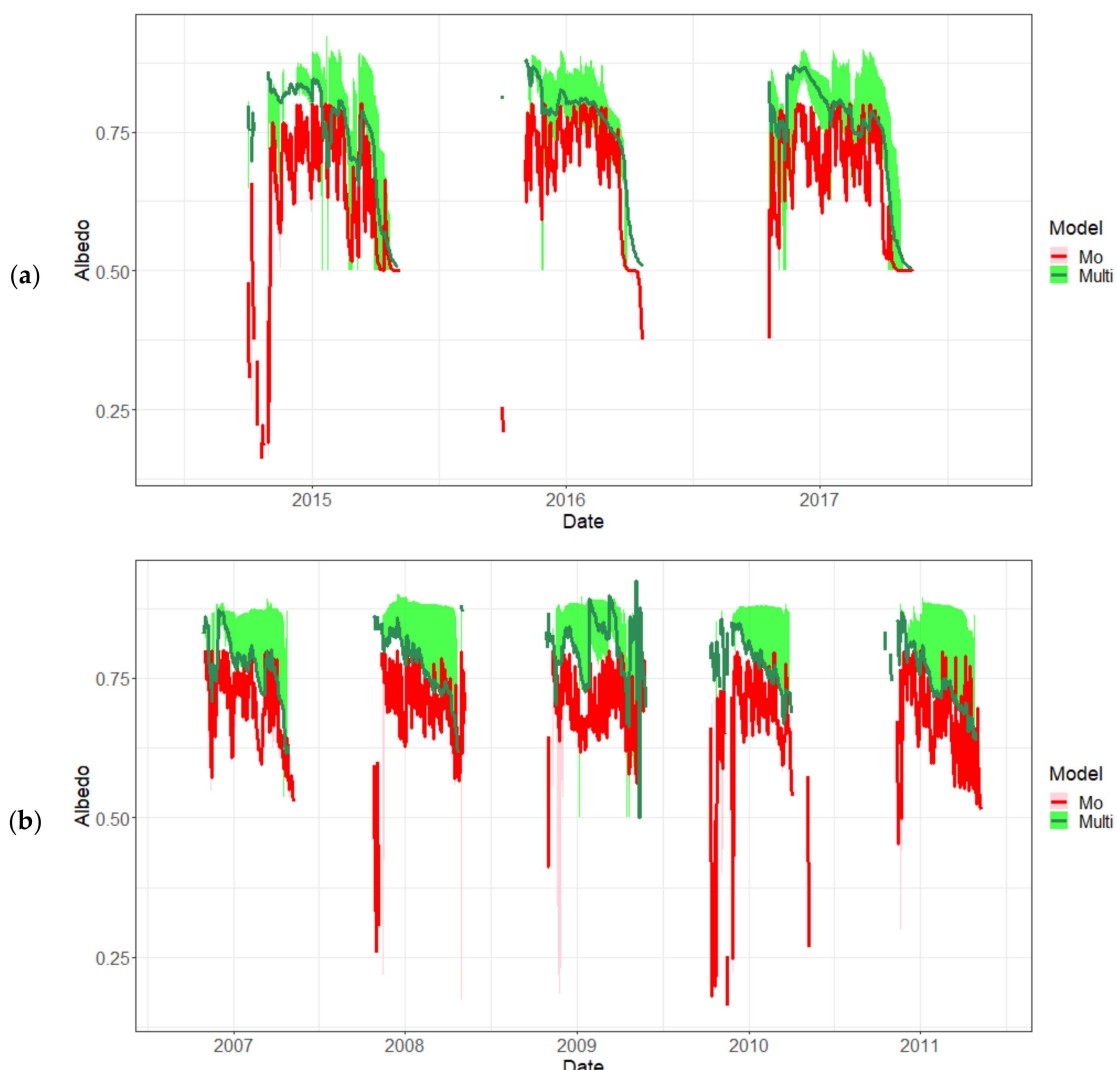

**Figure A5.** Albedo time series modeled by the top ten best sets of parameter values for the Mo model (pink envelop) and the Multi model (green envelop) for the (**a**) Lower Fantail and (**b**) Necopastic GMON stations. The best parameter sets are depicted by the red and green lines for the Mo and Multi models, respectively.

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
