# Peer review of "Extension of a Monolayer Energy-Budget Degree-Day Model to a Multilayer One"

_water, doi:10.3390/w16081089_

Round 1

Reviewer 1 Report

Comments and Suggestions for Authors

This manuscript presents an enhanced snowpack simulation through a multilayer modeling framework. While the contribution is significant, several concerns need addressing:

After utilizing iThenticate to check similarities, it has been identified that certain lines in the manuscript overlap with contents from a dissertation authored by one of the contributors. I recommend rephrasing or eliminating these duplicated sections to ensure originality.

The manuscript appears lengthy, and Table 1, in its current format, may not be suitable for publication. Consider revising the table to condense its contents. Additionally, numerous formulas present in the text could potentially be moved to Supplementary Material.

Some elements of the manuscript, such as Fig. 1, replicate content already published elsewhere (edoi: 10.3390/w12123449). Please modify these features to maintain originality.

To broaden the manuscript's appeal internationally, it would be beneficial for the authors to cite relevant works focusing on snowmelt contributions to water resources in mountainous regions. For instance, studies concerning the Andes (https://doi.org/10.1002/joc.4804), Alps (https://doi.org/10.1007/s10584-021-03027-x), Middle East mountains (https://doi.org/10.1016/j.jhydrol.2021.126858), Sierra Nevada (https://doi.org/10.1038/nclimate2809), and the Tibetan Plateau (https://doi.org/10.1002/2017JD026524) could significantly enhance the manuscript's relevance and appeal to a wider audience.

Reviewer 2 Report

Comments and Suggestions for Authors

The paper provides a clear explanation of how a monolayer snowpack model has been converted to a multi layer model. The results comparison is thorough and does indicates advantages of the multi layer model, but not as clear cut as anticipated. There is one element that is not discussed in the case study phase the soil/snowpack interaction. Soil temperature/snow temperature discontinuity typically leads to a unique snow layer near the soil horizon that the multi layer can better deal with. The authors either need to reflect on this in their paper if they had a basal snow layer, and if not comment on the potential inclusion and advantages of that for future research.

41: The references here are useful but neither is from a Canadian basin, add a reference from this region.

48-54: This section is out of place getting to details of snow chemistry and snow flakes, before the main issue of SWE and snowpack layers are discussed.  Can be removed or used later.

109:  Are GEOTOP and SeNORGE models limited for application across wide regions through time?

202: Figure 2 is not of good visual quality and the graphic design communicates poorly.

213: Can a layer be added at the base of the snowpack due to either formation of ice layer or a depth hoar layer?

301: Was this an oversight or a computational limitation?

407: You already used a Figure 2.

416: Would be useful to see a snow profile at each site acquired during study period that illustrates differences.

444: Explain in more detail what this diagram is depicting by thickness of the color bands.

515: After reading this section I do not see how the multi model demonstrated a clear improvement. What is the metric that determined this statement?

545: Figure 8 provides a good visual of how well each model fits observations, but it is difficult to compare. Figure 9 similar. Table 5 does relate the overall KGE values. It would be valuable to directly contrast the model performance in a single graph vs the paired graphs.

603: Is this difference consistently late or early for either model?

647: Is there any difference at the snow-ground interface between Mo and Multi?

690: Is this densest layer at the snow/ground interface a separate layer in Multi?

Comments on the Quality of English Language

No problems with english language.

Reviewer 3 Report

Comments and Suggestions for Authors

The manuscript is well-written and organized, although it contains several mistakes that need to be addressed. The following comments are provided:

1.      Modification is needed for Figure 1 as the upper line in "Snow on the ground?" is not displayed.

2.      There are two instances of Figure 2, appearing on page 7 and page 14. The numbering of figures should be adjusted accordingly.

3.      In Line 211, Line 297, Line 317, Lines 440-441, and Line 570, "Error! Reference source not found" is displayed.

4.      It is recommended to include future work at the conclusion of the "Conclusions" section.

5.      It is suggested to remove citations in the "Conclusions" section.

Reviewer 4 Report

Comments and Suggestions for Authors

The proposed article is based on the extension of a monolayer snow model of a hydrological model known as HYDROTEL to a multilayer whereby snow is a combination of ice and air. 

My comments are as follows:

1. The abstract needs to be more concise and clearly state the objective of the study and the result metrics. 

2. The novelty in the manuscript is not mentioned and what new is being added to the existing field. 

3. Please check line 211, 317, 440, 570 on missing reference. 

4. The methodology in the study is not made clear. 

5. The sensitivity analysis basis needs to be explained for figure 4 and 5.

6. The paper appears more to be a review paper on the available models, the authors need to present supporting arguments for this to be otherwise. 

7. The title has 'improving snowpack simulation to a multilayer one' has to be rephrased appropriately as it appears vague. How was this improved in the study?

Comments on the Quality of English Language

Fine, no major issues found. 

Round 2

Reviewer 1 Report

Comments and Suggestions for Authors

The raised comments have been properly addressed within the revision.

Reviewer 4 Report

Comments and Suggestions for Authors

The authors have addressed all issues mentioned appropriately hence I recommend accepting the paper.